# Ancient *Clostridium* DNA and variants of tetanus neurotoxins associated with human archaeological remains

Harold P. Hodgins[1], Pengsheng Chen[2,3], Briallen Lobb[1], Xin Wei[1], Benjamin J. M. Tremblay ®[1], Michael J. Mansfield ®[4], Victoria C. Y. Lee[1], Pyung-Gang Lee[2,3], Jeffrey Coffin[5], Ana T. Duggan ®[6], Alexis E. Dolphin[5], Gabriel Renaud ®[7] ✉, Min Dong ®[2,3] ✉ & Andrew C. Doxey ®[1] ✉

The analysis of microbial genomes from human archaeological samples offers a historic snapshot of ancient pathogens and provides insights into the origins of modern infectious diseases. Here, we analyze metagenomic datasets from 38 human archaeological samples and identify bacterial genomic sequences related to modern-day *Clostridium tetani*, which produces the tetanus neurotoxin (TeNT) and causes the disease tetanus. These genomic assemblies had varying levels of completeness, and a subset of them displayed hallmarks of ancient DNA damage. Phylogenetic analyses revealed known *C. tetani* clades as well as potentially new *Clostridium* lineages closely related to *C. tetani*. The genomic assemblies encode 13 TeNT variants with unique substitution profiles, including a subgroup of TeNT variants found exclusively in ancient samples from South America. We experimentally tested a TeNT variant selected from an ancient Chilean mummy sample and found that it induced tetanus muscle paralysis in mice, with potency comparable to modern TeNT. Thus, our ancient DNA analysis identifies DNA from neurotoxigenic *C. tetani* in archaeological human samples, and a novel variant of TeNT that can cause disease in mammals.

*Clostridium tetani*, the causative agent of the neuroparalytic disease tetanus, is an important bacterial pathogen of humans and animals. After its spores enter wounds, it germinates, diffuses through oxygen-depleted and necrotic tissue, and produces a highly potent neurotoxin (tetanus neurotoxin, TeNT) that paralyzes hosts[1,2] leading to spastic paralysis. This may present as a local or systemic effect, which can result in death due to paralysis of respiratory muscles and subsequent respiratory failure. As a wound-associated infectious disease, tetanus is estimated to have plagued *Homo sapiens* throughout history. Accounts of tetanus diagnostic features ("lockjaw") date back to written descriptions by Hippocrates (c. 380 BCE)[3,4] and the ancient Egyptians as seen in reports from the Edwin Smith papyrus (c. 1600 BCE)[5].

*C. tetani* was first isolated and cultivated in 1889[6] and an early isolate of *C. tetani* (the 1920 Harvard E88 strain) is still widely used as a reference today. Genome sequencing of the E88 reference strain revealed that it possesses a single ~2.8 Mb chromosome and a ~74-kb

[1]Department of Biology and the Waterloo Centre for Microbial Research, University of Waterloo, Waterloo, ON, Canada. [2]Department of Urology, Boston Children's Hospital, Boston, MA, USA. [3]Department of Surgery and Department of Microbiology, Harvard Medical School, Boston, MA, USA. [4]Genomics and Regulatory Systems Unit, Okinawa Institute of Science and Technology Graduate University, Onna, Okinawa, Japan. [5]Department of Anthropology, University of Waterloo, Waterloo, ON, Canada. [6]McMaster Ancient DNA Centre, Department of Anthropology, McMaster University, Hamilton, ON, Canada. [7]Department of Health Technology, Section of Bioinformatics, Technical University of Denmark, Kongens Lyngby, Denmark. ✉e-mail: gabre@dtu.dk; Min.Dong@childrens.harvard.edu; acdoxey@uwaterloo.ca

plasmid[7]. This genomic organization is largely maintained among known strains of *C. tetani*, with different strains varying in plasmid size[8–10]. The plasmid is critical to pathogenicity as it contains the key virulence genes including *tent* which encodes the neurotoxin and *colT* which encodes a collagenase enzyme involved in tissue degradation. Based on comparative genomic analysis, modern *C. tetani* strains cluster into two phylogenetically distinct clades[10], which are closely related and exhibit low genetic variation with average nucleotide identities of 96–99%. Similarly, the *tent* gene is extremely conserved and exhibits 98–100% nucleotide identity across all strains. Modern *C. tetani* genomes therefore offer a limited perspective on the true diversity of *C. tetani* and its evolutionary history as a human pathogen.

The sequencing and analysis of ancient DNA (aDNA) from archeological samples provides unprecedented access to ancestral genomic information, and insights into the origins and evolution of modern species. In addition to human DNA, a significant proportion of genetic material preserved within ancient specimens is of microbial origin[11–13]. Ancient microbial DNA, including that from ancient pathogens that once impacted humans, can be found within mummified remains, paleofeces, bones, and teeth (pulp and dental calculus)[14]. However, it is important to note that ancient human samples contain a mixture of endogenous DNA from ancient microbes as well as contaminant DNA (e.g., modern DNA) derived from environmental sources, human handling of samples, and issues with sample storage in museums and laboratories[15–17]. It has been estimated that endogenous DNA makes up less than 5% of extracted DNA from ancient samples, with the remaining 95% derived from exogenous microorganisms[13,18]. Distinguishing endogenous and exogenous microbial DNA represents an ongoing challenge for paleomicrobiology.

Although distinguishing ancient microbial DNA from modern contaminant DNA may be challenging, pioneering work in the ancient DNA field has revealed molecular signatures of DNA damage that help distinguish ancient DNA from modern DNA[19,20]. Specifically, damaged ancient DNA is associated with an increase rate of deaminated cytosine residues that accumulate near the ends of DNA molecules. Due to the misincorporation of thymine instead of uracil by polymerases during amplification, this results in an apparent increased frequency of C → T transitions at the beginning and G → A transitions at the ends of sequence fragments[19,20]. Damage levels exceeding 10% are considered to be indicative of genuinely ancient DNA[21]. However, damage levels also depend on whether the DNA extract has been treated with uracil-DNA-glycosylase (UDG), which destroys the signal in the case of full UDG treatment, or reduces the signal at the ends of DNA molecules in the case of partial UDG treatment[22]. Once authenticated, reconstructed ancient microbial genomes can be compared with modern strains to investigate the genomic ancestry and adaptations underlying the emergence of historical epidemic strains. Groundbreaking aDNA studies on the evolutionary origins and emergence of major infectious diseases have been carried out in recent years including studies of *Mycobacterium tuberculosis*[23], the plague bacterium *Yersinia pestis*[24], *Mycobacterium leprae*[25], *Helicobacter pylori*[26], hepatitis B[27], and variola virus[28].

Here, using a large-scale metagenomic data mining of millions of sequencing datasets, we report the discovery of novel *C. tetani*-related genomes including neurotoxin genes from datasets associated with human archeological samples. Some strains and neurotoxins are phylogenetically distinct from modern forms, and some strains show strong hallmarks of ancient DNA damage indicative of an ancient origin. We further demonstrate that a neurotoxin variant identified from a ~6000-year-old Chinchorro mummy aDNA sample produces tetanus-like paralysis in mice with a potency comparable to modern tetanus neurotoxins. Our findings uncover a widespread occurrence of *C. tetani* and related species associated with aDNA samples, expanding our understanding of the evolution and diversity of this important human pathogen.

## Results

### Identification and assembly of *C. tetani*-related genomes from aDNA samples

To explore the evolution and diversity of *C. tetani*, we performed a large-scale search of the entire NCBI Sequence Read Archive (SRA; 10,432,849 datasets from 291,458 studies totaling ~18 petabytes; June 8, 2021) for datasets potentially containing *C. tetani* DNA signatures. Since typical homology-based search methods (e.g., BLAST[29]) could not be applied at such a large scale, we used the recently developed Sequence Taxonomic Analysis Tool (STAT)[30] to search the SRA and identified 136 sequencing datasets possessing the highest total *C. tetani* DNA content [*k*-mer abundance >23,000 reads, *k* = 32 base pair fragments mapping to the *C. tetani* genome] (Fig. 1a and Supplementary Data 1). Our search identified 28 previously sequenced *C. tetani* genomes (which serve as positive controls), as well as 108 uncharacterized sequencing runs (79 of human origin) with high predicted levels of *C. tetani* DNA content. Unexpectedly, 76 (96.2%) of these are aDNA datasets collected from human archeological specimens (Fig. 1a), with the remaining three datasets being from modern human gut microbiome samples.

These 76 ancient DNA datasets are sequencing runs derived from 38 distinct archeological samples, which include tooth samples from aboriginal inhabitants of the Canary Islands from the 7th to 11th centuries CE[31], tooth samples from the Sanganji Shell Mound of the Jomon in Japan (~1044 BCE)[32], Egyptian mummy remains from ~1879 BCE to 53 CE[33], and ancient Chilean Chinchorro mummy remains from ~3889 BCE[34] (Supplementary Data 2). The 38 aDNA samples vary in terms of sample type (31 tooth, 6 bone and 1 chest extract), burial practices (27 regular inhumation and 11 mummies), sequencing method (26 shotgun datasets and 12 bait-capture approaches), and DNA treatment (6 UDG-treated, 5 partial UDG-treated and 27 untreated samples), all of which needs to be considered for interpretation of downstream analysis (Supplementary Data 2).

Although these archeological samples are of human origin, STAT analysis of the 38 DNA samples predicted a predominantly microbial composition (~90% median across samples, Supplementary Fig. 1). The predominance of microbial DNA in ancient human tooth samples is expected and consistent with previous studies which have shown microbial DNA proportions as high as 95–99%[13,17,18,35]. *C. tetani*-related DNA was consistently abundant among predicted microbial communities, detected at 13.8% average relative abundance (Supplementary Fig. 1 and Supplementary Data 3). A total of 85 species were detected at >= 2% abundance in at least one sample (Supplementary Data 4). While 65 of these species have been associated with humans or animals, 20 species have an environment-specific origin, and provide an estimate of possible environmental microbial contamination that could aid in interpretation of results (Supplementary Data 4, Supplementary Fig. 2). Putative environment-specific microbes make up a low proportion of the microbially classified reads at levels <=10% for 33 samples, and <=5% for 24 samples (Supplementary Data 5). The three samples with the highest estimated proportions of reads from putative environment-specific microbes were Tenerife-012-Tooth, Vác-Mummy-Tissue, and Tenerife-013-Tooth (Supplementary Data 5). Also noteworthy is that *M. tuberculosis* and *Y. pestis* were detected (Supplementary Fig. 1) in several datasets associated with bait-capture sequencing of *M. tuberculosis* and *Y. pestis* from archeological samples[36–38].

To further explore the putative *C. tetani* in aDNA samples, we performed metagenome assembly using MEGAHIT[39] for each individual sample and taxonomically classified assembled contigs using both Kaiju[40] and BLAST[29] to identify those mapping unambiguously to *C. tetani* and not other bacterial species (Supplementary Data 6 and 7). A majority (73%) of the alignments between assembled contigs and reference *C. tetani* genomes had percentage identities exceeding 99% (Fig. 1b). Ninety percent of the alignments had percentage identities

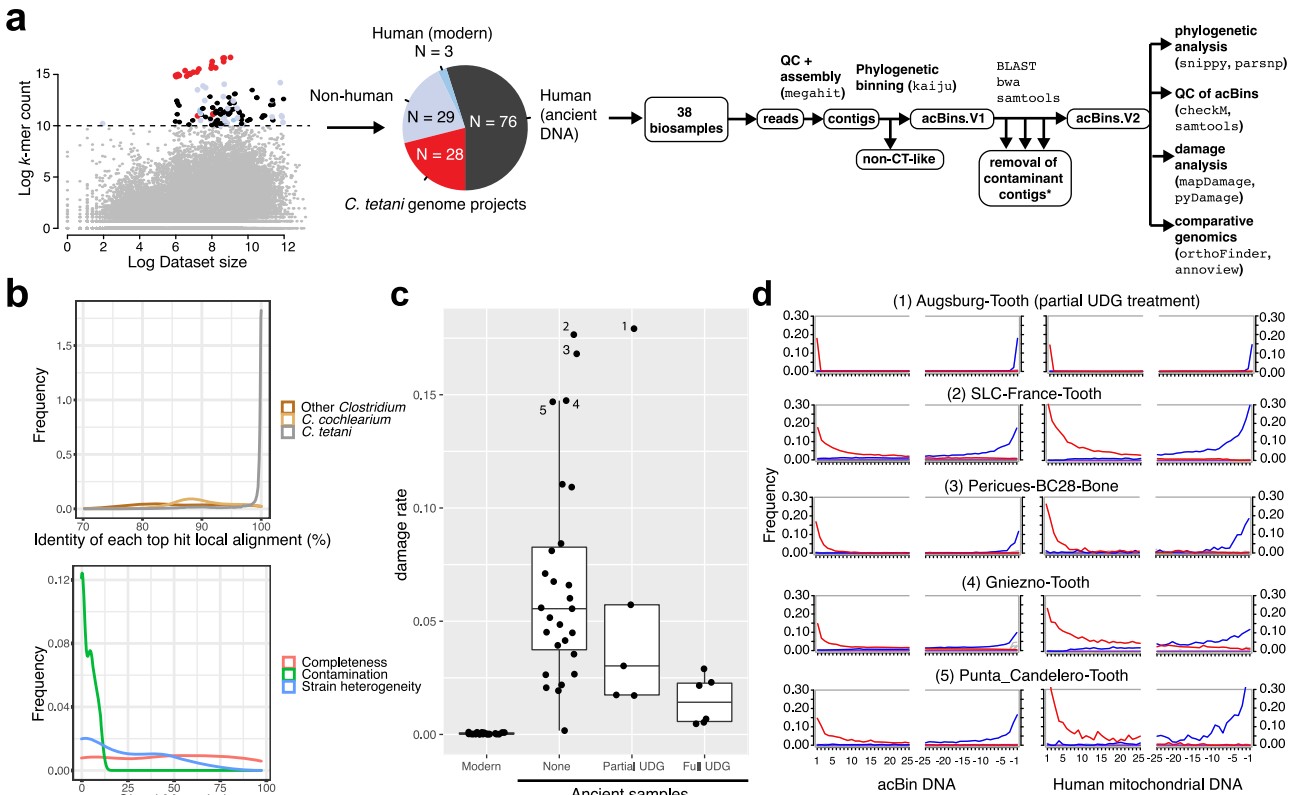

**Fig. 1 | Petabase-scale screen of the NCBI sequence read archive reveals *C. tetani*-related genomes in ancient human archeological samples. a** General bioinformatic workflow starting with the analysis of 43,620 samples from the NCBI sequence read archive. Each sample is depicted according to its *C. tetani* k-mer abundance (y axis) versus the natural log of the overall dataset size in megabases (x axis). A threshold was used to distinguish samples with high detected *C. tetani* DNA content, and these data points are colored by sample origin: modern *C. tetani* genomes (red), non-human (light blue), modern human (blue), ancient human (black). The pie chart displays a breakdown of identified SRA samples with a high abundance of *C. tetani* DNA signatures. The 38 aDNA samples predicted to contain *C. tetani* DNA were further analyzed as shown in the bioinformatic pipeline on the right. **b** Top−density plot of the percentage identities of all BLAST local alignments detected between acBins and reference genomes including *C. tetani*, *C. cochlearium*, and other *Clostridium* spp. Bottom−density plot of the checkM results for the

38 acBins including estimated completeness, contamination, and strain heterogeneity levels. Completeness and contamination levels are percentage values. **c** MapDamage damage rates (5′ C → T misincorporation frequency) for acBins (n = 38 biologically independent samples) subdivided by UDG treatment [none (n = 27), partial (n = 5), and full (n = 6)]. Also shown are the damage rates for modern *C. tetani* genomes (n = 21 biologically independent samples). The boxplots depict the lower quartile, median, and upper quartile of the data, with whiskers extending to 1.5 times the interquartile range (IQR) above the third quartile or below the first quartile. **d** Damage plots for the top five acBins with the highest damage rates, and corresponding mtDNA damage plots. Shown is the frequency of C → T (red) and G → A (blue) misincorporations at the first and last 25 bases of sequence fragments. Increased misincorporation frequency at the edges of reads is characteristic of ancient DNA. Source data for (**a**–**d**) are provided as a Source Data file.

exceeding 90%, suggesting that a large fraction of assembled contigs are highly similar to regions of modern *C. tetani* genomes. Based on mapping of reads to the *C. tetani* chromosome, the 38 samples had a 1× percent coverage ranging from 28 to 94% (mean of 78.3%) and a 5× coverage ranging from 9 to 93% (mean of 57.5%) (Supplementary Data 2). A subset of 16 samples had a 1× *C. tetani* chromosome coverage exceeding 90%.

For each of the ancient DNA samples, we binned together all *C. tetani*-like contigs to result in 38 putative, ancient DNA-associated clostridial genome bins or "acBins". We then performed QC analysis of each acBin using CheckM[41] to estimate genome completeness and contamination (Fig. 1b and Supplementary Data 8). CheckM estimates genome completeness based on the detected presence of taxon-specific marker genes, and uses duplicated marker genes (if present) to estimate contamination and heterogeneity[41]. Eighteen acBins were more than 50% complete and 11 were more than 70% complete. Thirty-seven acBins had low (<10%) checkM contamination (Supplementary Data 8). acBins with higher genome completeness were associated with datasets produced by shotgun sequencing rather than capture methods, as these datasets had higher levels of *C. tetani* DNA content (Supplementary Fig. 3). We also examined the acBins for potential

strain heterogeneity using two independent approaches: CheckM estimation (Supplementary Data 8) as well as quantification of per-base heterogeneity from mapped reads (Supplementary Data 2). These two metrics had a weak but significant correlation (r = 0.38, P = 0.019) (Supplementary Fig. 4a). Five strains (Sanganji-A2-Tooth, Chinchorro-Mummy-Bone, SLC-France-Tooth, Karolva-Tooth, Chincha-UC12-24-Tooth) were identified as possessing higher estimated levels of strain variation, but all were below 6% (CheckM) and 1.1% (average base heterogeneity).

## A subset of *C. tetani* genomes from archeological samples are of ancient origin

Using the tools MapDamage2[42] and pyDamage[43], we then examined the 38 acBins for elevated C → T misincorporation rates at the ends of molecules, a characteristic pattern of aDNA damage[19,20]. Since these patterns are known to be affected by UDG treatment, we examined damage rates separately for full UDG, partial UDG, and untreated samples (Fig. 1c). As expected, we observed the highest damage rates in the untreated samples, and the lowest damage rates in the full UDG-treated samples, indicating that the damage rates have been suppressed in some samples by UDG treatment. The damage rates

calculated by MapDamage and PyDamage were highly similar with a Pearson correlation of $r = 0.99$ (Supplementary Data 2). Damage plots for all samples are shown in Supplementary Fig. 5 with additional data available in Supplementary Data 9 and 10.

Overall, seven acBins possessed a damage rate (5' C→T misincorporation rate) exceeding 10%, which is indicative of aDNA[21] (top 5 shown in Fig. 1d). In addition, all of the acBins except one ("Chincha-UC12-12-Tooth") were verified by pyDamage as containing ancient contigs with $q$ values < 0.01 (Supplementary Data 10). The highest damage rate (17.9%) occurred in the acBin from the "Augsburg-Tooth" sample, which is the third oldest sample in our dataset (~2253 BCE), despite this sample being partially UDG-treated (Fig. 1d). As controls, evidence of ancient DNA damage was also observed in the corresponding human mitochondrial DNA (mtDNA) from the same ancient samples (Supplementary Fig. 5 and Supplementary Data 2), but not for modern *C. tetani* samples (Fig. 1c). In addition, no damage was detected in the three human gut-derived *C. tetani* bins identified by our screen.

In general, we observed a significant correlation between damage rates of acBin DNA and corresponding human mtDNA from the same sample ($R^2 = 0.38$, $P = 2.8E\text{-}03$, two-sided Pearson) (Supplementary Fig. 6). However, acBin damage rates were generally lower than the corresponding human mtDNA rates, especially for some samples (e.g., Tenerife-004, Tenerife-013, Chinchorro-Mummy-Bone) (Supplementary Figs. 5 and 6), which may suggest that a subset of the archeological samples have been colonized by *C. tetani* at later dates (see "Discussion"). Damage rates were higher for noncapture datasets as these generally received no UDG treatment (Supplementary Fig. 7a), and higher for samples associated with regular inhumations than those from mummies (Supplementary Fig. 7b). We also observed a significant correlation between acBin damage level and sample age, but only for mummy-derived samples ($R^2 = 0.50$, $P = 0.014$) (Supplementary Fig. 7c). Together, these data suggest that a subset of the acBins display evidence of ancient DNA damage and are plausibly of an ancient origin.

### Identification of novel *C. tetani* lineages and a potentially new *Clostridium* species from ancient samples

To explore the phylogenetic relationships between the acBins and modern *C. tetani* strains, we first aligned their contigs to the reference *C. tetani* genome along with 41 existing, non-redundant *C. tetani* genomes[10], and clustered the genomes to produce a dendrogram (Fig. 2a). Five acBins were omitted due to extremely low (<1%) genome coverage (see "Methods"), which could result in phylogenetic artifacts. We also included *C. cochlearium* as an outgroup, as it is the closest known related species to *C. tetani* based on phylogenomic analysis of available genomes[44,45]. Assessment of the genome-wide alignment for potential recombination showed no difference in estimated recombination levels for acBins compared to modern *C. tetani* genomes (Supplementary Fig. 8).

The genome-based dendrogram of the acBins and modern *C. tetani* strains (Fig. 2a) matches the expected phylogenetic structure and contains all previously established *C. tetani* lineages[10]. Ultimately, the acBins can be subdivided into those that cluster clearly within existing *C. tetani* lineages 1 or 2 and those that do not, which we have labeled "X" (8 acBins) and "Y" (1 acBin). Visualization of the acBin samples on the world map revealed a tendency for geographical clustering among acBins from the same phylogenetic lineage (Fig. 2b). For example, lineage 1H acBins originate from ancient samples collected in the Americas, whereas most lineage 2 acBins originate outside of the Americas, and most clade X samples originate in Europe (Fig. 2b). Interestingly, some samples from the same region (e.g., Canary Island samples, and Egyptian samples) contain diverse *C. tetani* lineages, which may be influenced by several factors (see "Discussion").

**acBins from *C. tetani* lineages 1 and 2.** Twenty-four acBins fall within the *C. tetani* tree and possess average nucleotide identities (ANIs) of 96.4% to 99.7% to the E88 reference genome (Supplementary Data 2), which is within the range considered to be the same species[46]. These include new members of clades 1B (1 acBin), 1 F (1 acBin), 1H (9 acBins), and 2 (9 acBins), expanding the known genomic diversity of clade 1H which previously contained a single strain and clade 2 which previously contained five strains (Fig. 2a). Four additional acBins clustered generally within clade 1 but outside of established sublineages (Fig. 2a).

In addition, we used Parsnp[47] to construct a more stringent, core SNP-based phylogeny from a reduced set of 11 acBins that aligned to the reference *C. tetani* genome and passed several criteria (see "Methods") (Fig. 2c and Supplementary Fig. 9). Only acBins from established *C. tetani* lineages 1 and 2 passed these criteria, and their phylogenetic positioning is consistent with their clustering pattern (Fig. 2a). The reads associated with the core SNP alignment also showed reduced per-base heterogeneity when mapped to contigs (Supplementary Fig. 4b). Notably, acBins from the Sanganji, Tenerife, Chinchorro, and Chincha samples do not show evidence of branch shortening in the tree indicative of ancient genomes, and instead cluster with modern strains. These acBins tend to have higher rates of strain variation, which could affect branch lengths, or low damage rates potentially indicative of a more recent origin (Supplementary Data 2).

We also assembled a novel strain of *C. tetani* from a human gut sample (SRR10479805) which phylogenetically clustered with strain NCTC539 (98.7% average nucleotide identity; Supplementary Data 11) from lineage 1 G. The other two identified human gut samples were removed from further analysis as they predominantly matched *C. cochlearium* based on BLAST analysis.

**acBins from clade "X" and branch "Y".** Nine acBins clustered outside of the *C. tetani* species clade. Eight of these cluster together as part of a divergent clade (labeled "X") (Fig. 2a). These samples span a large timeframe from ~2290 BCE to 1787 CE, are predominantly (7 of 8) of European origin (Fig. 2b and Supplementary Fig. 10), and come from variable burial contexts including single cave burials, cemeteries, mass graves and burial pits[37,48–53] (Supplementary Data 2). Two of the samples from sites in Latvia and France are from plague (*Y. pestis*) victims[37,53], and another is from an individual with tuberculosis[38]. The highest quality clade X acBin is from sample "Augsburg-Tooth" (~2253 BCE), with 53.9% estimated completeness and 4.11% contamination (Supplementary Data 8). Comparison of clade X acBins to other *Clostridium* species revealed that they are closer to *C. tetani* and *C. cochlearium* than any other *Clostridium* species available in the existing NCBI database, but are divergent enough to be considered a distinct species. On average, based on fastANI[54] analysis of orthologous sequences[54] Clade X genomes have 86.5 + − 1.7% ANI to *C. tetani* strain E88, and 85.1 + − 1.3% ANI to *C. cochlearium* (Supplementary Fig. 11a and Supplementary Data 12). Based on ANI analysis of the whole genome alignment, clade X genomes have 90.8 + − 0.22% ANI to strain E88 (Supplementary Data 2). These similarities were confirmed by analysis of BLAST alignment identities between clade X contigs and reference genomes (Supplementary Fig. 11b). As in the genome-wide tree, individual marker genes (*rpsL*, *rpsG*, and *recA*) from clade X acBins also clustered as divergent branches distinct from *C. tetani* and *C. cochlearium* (Supplementary Figs. 12–14). Finally, we re-examined the damage patterns according to phylogenetic clade, and found that clade X genomes possess the highest mean damage; 6/8 clade X genomes have a damage level exceeding 5% and 3/8 exceed 10% (Supplementary Fig. 7d and Supplementary Data 2). These analyses suggest that clade X may represent a previously unidentified lineage of *Clostridium*, including members of ancient origin. We designated this group *Clostridium* sp. X.

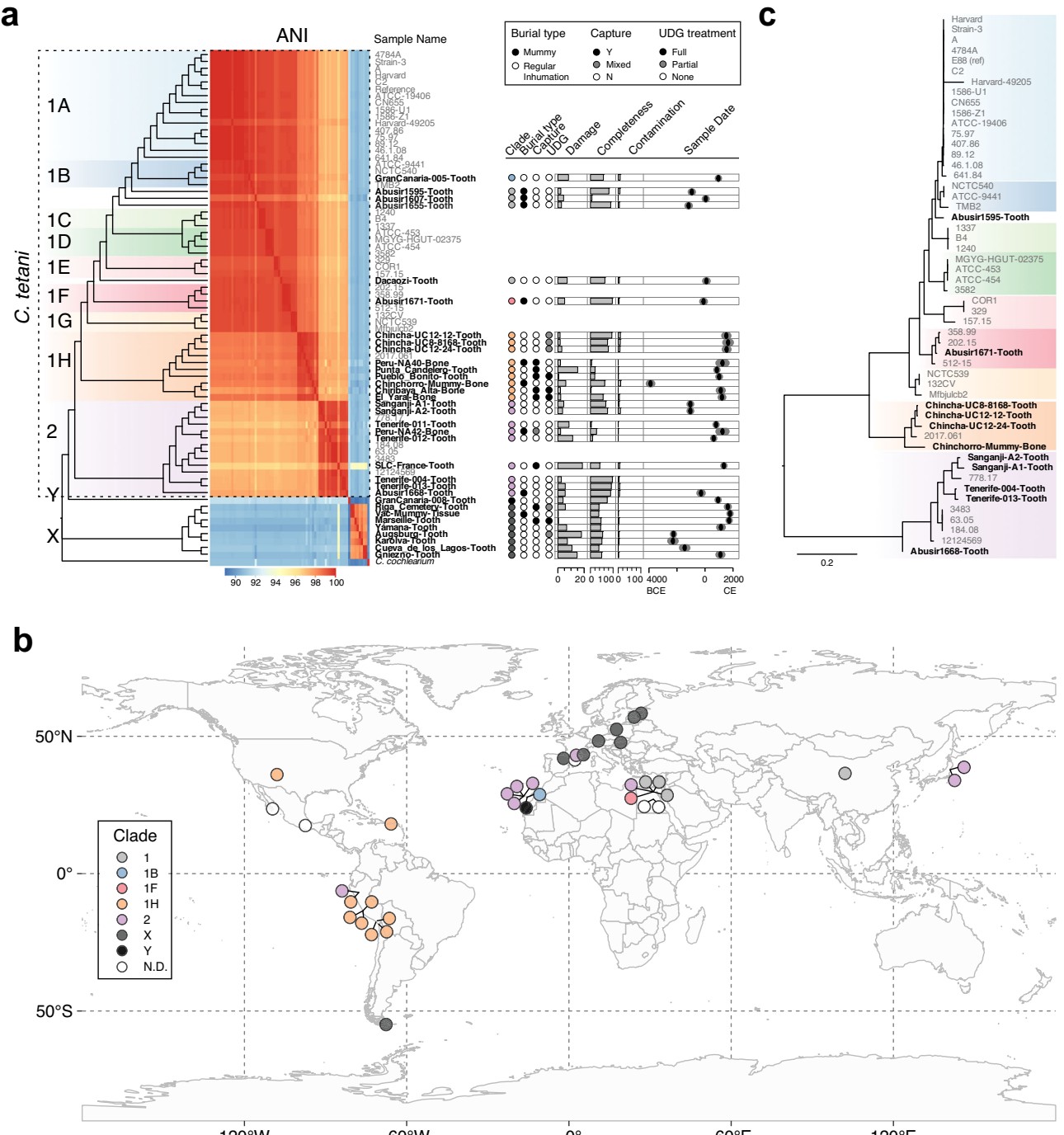

**Fig. 2 | Phylogenetic analysis reveals known and novel lineages of *C. tetani* in ancient DNA samples, as well as a previously unidentified *Clostridium* species ("X").** **a** Dendrogram depicting relationships of acBins from ancient samples with modern *C. tetani* genomes. Novel branches are labeled "X" and "Y", which are phylogenetically distinct from existing *C. tetani* genomes. Shown on the right of the dendrogram are metadata and statistics associated with each acBin including the estimated date of the associated archeological sample. All metadata can be found in Supplementary Data 2. **b** Geographic distribution of ancient DNA samples from

which the 38 acBins were identified. Each sample is colored based on the acBin clustering pattern shown in (**a**). The global map was derived from the Natural Earth [https://www.naturalearthdata.com/] medium-scale data and plotted using the rnaturalearth and ggplot2 R packages. **c** SNP-based phylogenetic tree of a subset of acBins from lineage 1 and 2 showing high similarity and coverage to the *C. tetani* reference genome. See Supplementary Fig. 9 for more details. Source data for (**a**, **c**) are provided as a Source Data file.

One sample ("GranCanaria-008-Tooth" from the Canary Islands dated to ~935 CE) also formed a single divergent branch (labeled "Y") clustering outside all other *C. tetani* genomes (Fig. 2a). Based on CheckM analysis, this acBin is of moderate quality with 74% completeness, and 0.47% contamination (Supplementary Data 8). A comparison of the GranCanaria-008-Tooth acBin to the NCBI genome

database revealed that it is closely related to *C. tetani* and more distant to other available *Clostridium* genomes (Supplementary Data 13). Based on fastANI[54], it exhibits an ANI of 87.3% to *C. tetani* E88, and 85.1% to *C. cochlearium*, below the 95% threshold typically used for species assignment (Supplementary Data 13). Based on ANI analysis of the whole genome alignment, it has a 91.2% ANI to strain E88

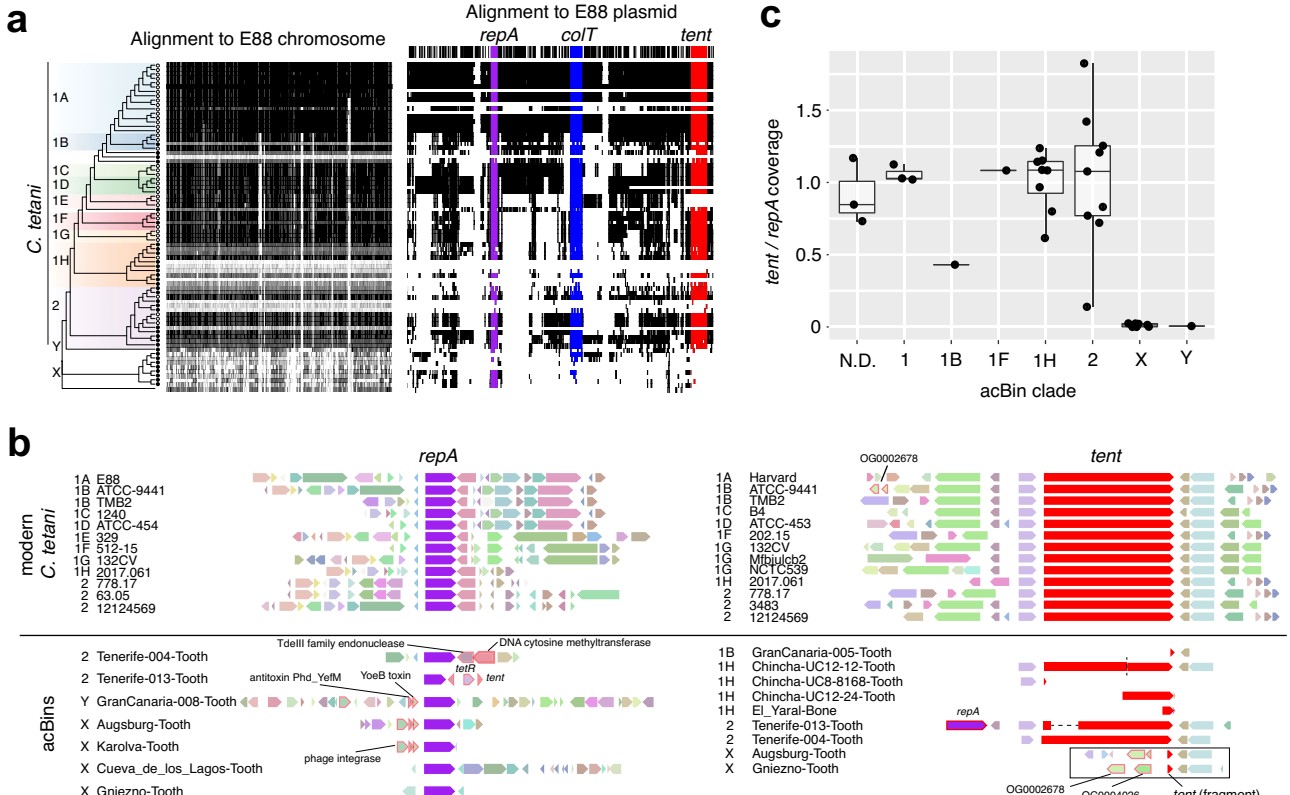

**Fig. 3 | Comparative genomics of acBins versus modern *C. tetani* strains.**
**a** Visualization of the chromosomal and plasmid multiple sequence alignment. Orthologous blocks are shown in black and the missing sequence is colored white. The reference gene locations are plotted above the alignments. **b** Gene neighborhoods surrounding the *repA* gene (left) and *tent* gene (right) in modern strains versus acBins. Selected unique differences identified in acBin gene neighborhoods are highlighted. The boxed region shows the assembled *tent* locus in two clade X acBins. Comparison reveals a putative deletion event in the clade X strains that has removed the majority of the *tent* gene along with five upstream genes, leaving behind conserved flanking regions. See Supplementary Fig. 18 for more information. **c** Per-clade coverage of the *tent* gene normalized to the coverage of *repA*. The data include *n* = 33 biologically independent samples, including acBins from clade 1 (*n* = 3), 1B (*n* = 1), 1 F (*n* = 1), 1H (*n* = 8), 2 (*n* = 9), X (*n* = 7), Y (*n* = 1), and acBins whose clade affiliation could not be determined (N.D., *n* = 3). The coverage was calculated as the average depth of coverage based on mapped reads to each gene. The boxplots depict the lower quartile, median, and upper quartile of the data, with whiskers extending to 1.5 times the interquartile range (IQR) above the third quartile or below the first quartile. See Supplementary Fig. 17 for the associated read pileups. Source data for (**a**–**c**) are provided as a Source Data file.

(Supplementary Data 2). To further investigate the phylogenetic position of this species, we built gene-based phylogenies with ribosomal marker genes *rpsL, rpsG* and *recA* (see Supplementary Figs. 12–14). Each of these three genes support the GranCanaria-008-Tooth lineage as a divergent species distinct from *C. tetani*. The damage level for this acBin is relatively low (~4.0%), whereas its human mtDNA damage level is ~11.6% (Supplementary Fig. 5). We designated this acBin *Clostridium* sp. Y.

**Genomic similarities and differences in *C. tetani*-related strains from ancient samples**
We next carried out a comprehensive comparison of genome content and structure between the acBins and modern *C. tetani* strains. We first clustered protein-coding sequences from all modern genomes and acBins into a set of 3729 orthologous groups, and compared their presence/absence across all strains (see "Methods" and Supplementary Data 14). Based on this analysis, we observed considerable overlap in gene content between the acBins versus the modern reference genomes, with the greatest overlap observed between acBins from *C. tetani* lineages (1 and 2) and the smallest overlap observed for *Clostridium* sp. X (Supplementary Fig. 11c). For instance, plasmid genes from the E88 reference genome were on average detected in 61% of the most complete acBins from Fig. 1c (comparable to 69% in modern *C. tetani* genomes), and only 35% of other acBins (Supplementary Data 15). Twenty orthogroups from the E88 plasmid were found in all

of these acBins, including the plasmid-specific genes *repA*, *colT*, and *tent* (Supplementary Data 15). In addition to these genes, sporulation-related genes are also highly conserved across the most complete acBins. Of 80 identified sporulation-related genes present in strain E88, 52 of these were detected in 100% of the most complete acBins, and 69/80 were present at over 90% frequency (Supplementary Data 16). Thus, we conclude that key *C. tetani* functions, including plasmid replication, collagen degradation, neurotoxin production, and sporulation, are conserved in a subset of acBins (i.e., those in Fig. 1c) for which enough genomic data was available to assemble genomes with moderate-high completeness.

We then examined genome similarities by visualizing the alignment of each genome to the reference E88 chromosome and plasmid (Fig. 3a). Several low-coverage acBins can be seen in *C. tetani* lineages 1 and 2 (Fig. 3a), which is expected given their low completeness estimates (Fig. 2a). However, the divergent GranCanaria-008-Tooth genome (branch "Y") and *Clostridium* sp. X consistently have a low alignment coverage, similar to that of *C. cochlearium* (Fig. 3a), which we suspected may be due in part to these species being more distantly related to *C. tetani*. Consistent with the idea that clade X represents a distinct species from *C. tetani*, we identified fourteen genes present in four or more clade X members and absent from all other *C. tetani* genomes. The genomic context of four of these genes (labeled by orthogroup) is shown in Supplementary Fig. 15. Although these genes are unique to clade X, their surrounding genes are conserved in other

*C. tetani* genomes, implying that genome rearrangements may have resulted in these genes being either gained in *Clostridium* sp. X or lost in *C. tetani*.

To examine differences in plasmid gene content and structure directly, we then compared the gene neighborhoods surrounding the plasmid-marker genes *repA* and *colT* (Fig. 3b, expanded data shown in Supplementary Fig. 16). In several acBins from *C. tetani* lineages 1 or 2, the gene neighborhoods surrounding these genes are similar to that in modern strains (Supplementary Fig. 16). However, particularly in *Clostridium* sp. X and Y, we identified unique gene clusters distinct from those in modern strains. For example, in two *Clostridium* sp. X genomes and the *Clostridium* sp. Y genome, we identified a conserved toxin/antitoxin pair and a phage integrase flanking the *repA* gene (Fig. 3b). In *Clostridium* sp. Y, these genes were found on an assembled 53.6 kb contig ("SAMEA104281224_k141_98912"), which was indeed predicted as a plasmid by the RFplasmid program with a 70.4% vote using the *Clostridium* model[55]. We also observed a unique gene arrangement surrounding *colT* that is conserved in two clade X genomes (Supplementary Fig. 16). Additional differences were identified in a few lineage 2 acBins; for example, Tenerife-004-Tooth contains unique genes neighboring *repA*, and the Tenerife-013-Tooth acBin uniquely encodes the *repA* gene adjacent to its *tent* and *tetR* gene (Fig. 3b).

We then performed a detailed comparison of the plasmid-encoded neurotoxin gene, *tent*, and its gene neighborhood (where possible) across the strains. As shown in Fig. 3a as well as based on mapped read coverage to these regions (Fig. 3c, Supplementary Fig. 17, and Supplementary Data 17), the *tent* gene was detected at a relatively high depth of coverage in acBins from *C. tetani* lineages 1 and 2. The *tent* gene neighborhood structure from lineage 1 or 2 acBin strains is also similar or identical to that in modern strains, with the exception of Tenerife-013-Tooth (as it encodes the *repA* gene nearby) (Fig. 3b).

However, in the acBins from lineage X and Y, the *tent* gene was either missing or was fragmented, suggesting a possible gene loss or pseudogenization event (Fig. 3c). This pattern can be seen clearly in read coverage plots (Supplementary Fig. 17) and when normalizing *tent* depth of coverage to that of the plasmid-marker gene, *repA* (Fig. 3c). The *tent* locus in the two *Clostridium* sp. X genomes for which assembly data is available over this region appears to have undergone a deletion event resulting in the deletion of over 90% of the *tent* sequence as well as 3 neighboring genes (Fig. 3b and Supplementary Fig. 18). This analysis further supports the idea that the *tent* fragment may be a nonfunctional pseudogene in these clade X strains.

Ultimately, our comparative genomic analysis of gene content and neighborhood structure demonstrates that the plasmids in several of the ancient samples (particularly those of *Clostridium* sp. X) are distinct from modern *C. tetani* plasmids, while the plasmids of acBins from lineages 1 and 2 are similar to those of existing *C. tetani* strains. This reinforces our earlier phylogenetic analysis indicating that clade X and branch Y represent new *Clostridium* species that are closely related to but distinct from *C. tetani*.

### Identification and experimental testing of novel TeNT variants

Given the considerable scientific and biomedical importance of clostridial neurotoxins, we next focused on *tent* and reconstructed a total of 18 *tent* gene sequences (all from lineage 1 and 2 acBins) from aDNA using a sensitive variant calling pipeline (see "Methods"). Six *tent* sequences have complete coverage, and 12 have 75-99.9% coverage (Supplementary Data 18). Six partial *tent* sequences were also reconstructed but had lower average depth of coverage as shown in the read pileups (Supplementary Fig. 17). Four of the reconstructed *tent* sequences are identical to modern *tent* sequences, while 14 (including two identical sequences) are novel *tent* variants with 99.1–99.9%

nucleotide identity to modern *tent*, comparable to the variation seen among modern *tent* genes (98.6–100%). We then built a phylogeny including the 18 *tent* genes from aDNA and all 12 modern *tent* sequences (Fig. 4a). The *tent* genes clustered into three subgroups with modern and aDNA-associated *tent* genes found in subgroups 1 and 2, and aDNA-associated *tent* genes forming a novel subgroup 3 (Fig. 4a). All three of the *tent* sequences in the novel *tent* subgroup 3 are from clade 1H aDNA strains.

We then visualized the uniqueness of aDNA-associated *tent* genes by mapping nucleotide substitutions onto the phylogeny (Fig. 4b and Supplementary Fig. 19), and focusing on "unique" *tent* substitutions found only in ancient samples and not in modern *tent* sequences. We identified a total of 46 such substitutions that are completely unique to one or more aDNA-associated *tent* genes (Fig. 4b, Supplementary Fig. 20, and Supplementary Data 19), which were statistically supported by the stringent variant calling pipeline (Supplementary Data 20). The largest number of unique substitutions occurred in *tent*/Chinchorro from *tent* subgroup 3, which is the oldest sample in our dataset ("Chinchorro mummy bone", -3889 BCE). *tent*/Chinchorro possesses 18 unique substitutions not found in modern *tent*, and 12 of these are shared with *tent*/El-Yaral and 10 with *tent*/Chiribaya (Fig. 4b). The three associated acBins also cluster as neighbors in the phylogenomic tree (Fig. 2a), and the three associated archeological samples originate from a similar geographic region in Peru and Chile (Supplementary Fig. 21). These shared patterns suggest a common evolutionary origin for these *C. tetani* strains and their unique neurotoxin genes and highlight *tent* subgroup 3 as a distinct group of *tent* variants exclusive to ancient samples (Fig. 4a).

We then focused on *tent*/Chinchorro as a representative sequence of this group as its full-length gene sequence could be completely assembled. The 18 unique substitutions present in the *tent*/Chinchorro gene result in 12 unique amino acid substitutions, absent from modern TeNT protein sequences (L140S, E141K, P144T, S145N, A147T, T148P, T149I, P445T, P531Q, V653I, V806I, H924R) (Supplementary Data 21). Seven of these substitutions are spatially clustered within a surface loop on the TeNT structure[56] and represent a potential mutation "hot spot" (Fig. 4c). Interestingly, 7/12 amino acid substitutions found in TeNT/Chinchorro are also shared with TeNT/El-Yaral and 5/12 are shared with TeNT/Chiribaya (Supplementary Data 21). As highlighted in Fig. 4c, TeNT/Chinchorro and TeNT/El-Yaral share a divergent 9-aa segment (amino acids 141–149 in TeNT, P04958) that is distinct from all other TeNT sequences. Reads mapping to the *tent*/Chinchorro gene show a low damage level similar to that seen in the *C. tetani* contigs from this sample, and their damage pattern is weaker than the corresponding damage pattern from the associated human mitochondrial DNA (Fig. 4d).

Given the phylogenetic novelty and unique pattern of substitutions observed for the *tent*/Chinchorro gene, we sought to determine whether it encodes an active tetanus neurotoxin. For biosafety reasons, we avoided the production of a *tent*/Chinchorro gene construct and instead used sortase-mediated ligation to produce limited quantities of full-length protein toxin (Supplementary Fig. 22), as done previously for other neurotoxins[57,58]. This involved producing two recombinant proteins in *E. coli*, one constituting the N-terminal fragment and another containing the C-terminal fragment of TeNT/Chinchorro, and then ligating these together using sortase. The resulting full-length TeNT/Chinchorro protein cleaved the canonical TeNT substrate, VAMP2, in cultured rat cortical neurons (Fig. 4e), and can be neutralized with anti-TeNT anti-sera (Supplementary Fig. 22). TeNT/Chinchorro induced spastic paralysis in vivo in mice when injected to the hind leg muscle, which displayed a classic tetanus-like phenotype identical to that seen for wild-type TeNT (Fig. 4f). Quantification of muscle rigidity following TeNT and TeNT/Chinchorro exposure demonstrated that TeNT/Chinchorro exhibits a potency that is indistinguishable from TeNT (Fig. 4g). Together, these data

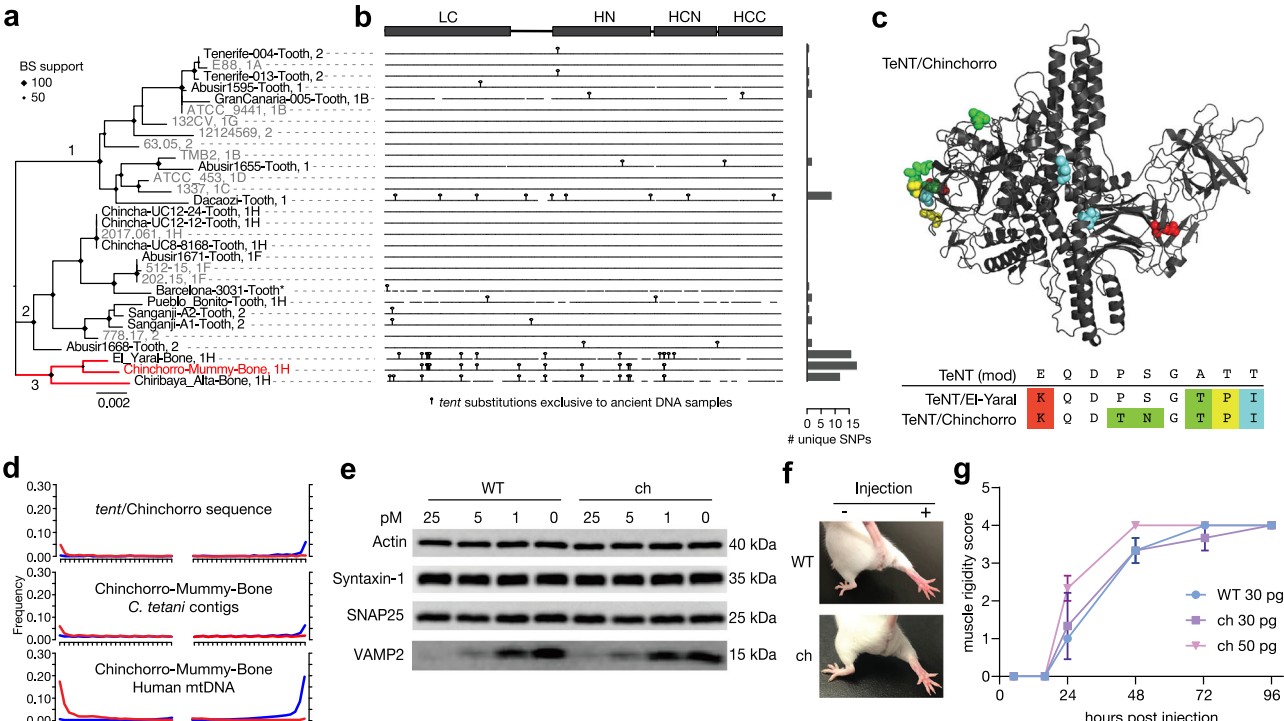

**Fig. 4 | Analysis and experimental testing of a novel TeNT lineage identified from ancient DNA. a** Maximum-likelihood phylogenetic tree of *tent* genes including novel *tent* sequences assembled from ancient DNA samples and a non-redundant set of *tent* sequences from existing strains in which duplicates have been removed (see "Methods" for details). The phylogeny has been subdivided into three subgroups. Sequences are labeled according to sample followed by their associated clade in the genome-based tree (Fig. 2a), except for the Barcelona-3031-Tooth sequence (*) as it fell below the coverage threshold. **b** Visualization of *tent* sequence variation, with vertical bars representing nucleotide substitutions found uniquely in *tent* sequences from ancient DNA samples. On the right, a barplot is shown that indicates the number of unique substitutions found in each sequence, highlighting the uniqueness of subgroup 3. **c** Structural model of TeNT/Chinchorro indicating all of its unique amino acid substitutions, which are not observed in modern TeNT sequences. Also shown is a segment of the translated alignment for a specific N-terminal region of the TeNT protein (residues 141–149, Uniprot ID P04958).

This sub-alignment illustrates a segment containing a high density of unique amino acid substitutions, four of which are shared in TeNT/El-Yaral and TeNT/Chinchorro. **d** MapDamage analysis of the *tent*/Chinchorro gene, and associated *C. tetani* contigs and mtDNA from the Chinchorro-Mummy-Bone sample. **e** Cultured rat cortical neurons were exposed to full-length toxins in culture medium at the indicated concentration for 12 h. Cell lysates were analyzed by immunoblot, and the image shown is a representative of four independent experiments. WT TeNT (uniprot accession # P04958) and TeNT/Chinchorro ("ch") showed similar levels of activity in cleaving VAMP2 in neurons. **f**, **g** Full-length toxins ligated by sortase reaction were injected into the gastrocnemius muscles of the right hind limb of mice. The extent of muscle rigidity was monitored and scored for 4 days (means ± s.e.; *n* = 3 per group, 9 total). TeNT/Chinchorro ("ch") induced typical spastic paralysis and showed a potency similar to WT TeNT. Source data for (**a**, **b**, **d**, **e**, **g**) are provided as a Source Data file.

demonstrate that the reconstructed *tent*/Chinchorro gene encodes an active and highly potent TeNT variant.

## Discussion

In this work, large-scale data mining of millions of existing genomic datasets revealed widespread occurrence of neurotoxigenic *C. tetani* and related species of *Clostridium* in aDNA samples from human archeological remains. Our study provides three main findings: (1) the first identification of neurotoxigenic *C. tetani* from archeological samples including several *C. tetani* strains of plausibly ancient origin; (2) the discovery of novel lineages of *C. tetani* as well as potentially new lineages of *Clostridium* (e.g., clade X and lineage Y); and (3) the identification of novel variants of TeNT including TeNT/Chinchorro which we demonstrate to be an active neurotoxin with extreme potency comparable to modern TeNT variants.

Our work is unique from previous studies of aDNA in several respects. First, using recent advances in petabase-scale genomic data mining[30], we were able to perform a large-scale survey of sequencing datasets including all publicly available DNA samples in the NCBI SRA, greatly enhancing our ability to discover patterns across spatially and temporally diverse datasets. Because of the massive size of the NCBI SRA database, traditional sequence-alignment-based methods such as BLAST are not computationally feasible. The STAT method[30] provided

a heuristic approach to identify potential datasets from the SRA that could then be targeted for deeper analysis including metagenomic assembly and phylogenomics. Importantly, we did not specifically look for *C. tetani* in ancient DNA, but rather this came as an unexpected finding from the results of our large-scale screen. Also unexpected was the considerable diversity of ancient samples in which we identified neurotoxigenic *C. tetani*, which revealed a strong association between this organism (and related species) and human archeological samples. Despite the abundance of environmental (e.g., soil metagenomic) samples in the SRA, these samples did not come to the surface of our genomic screen for *C. tetani*. This is consistent with the idea that, although *C. tetani* spores may be ubiquitous in terrestrial environments such as soil, these spores may be rare and so *C. tetani* DNA may not regularly appear at appreciable levels in shotgun metagenomes. Lastly, it is also noteworthy that the majority (31/38) of *C. tetani* aDNA samples in our study originated from teeth. Teeth are commonly used in aDNA studies due to the survival and concentration of endogenous aDNA content[59]. The colonization of tooth and bone tissue by *C. tetani* is also not unprecedented, as infections of these tissues have been previously but infrequently reported in clinical cases[60,61].

It is important to point out that the identification of neurotoxigenic *C. tetani* in aDNA samples alone is not sufficient to implicate tetanus as a cause of death or even suggest that the corresponding *C.*

*tetani* strains are contemporaneous with the archeological samples. *C. tetani* is a spore-forming organism and spores may exist for long periods[62] while providing protection from DNA damage[63]. Therefore, these factors may affect the damage profiles of *Clostridium*-derived DNA in ancient samples and limit our ability to distinguish modern strains from ancient ones that exist in the form of spores. More importantly, a variety of environmental factors and mechanisms may account for the presence of toxigenic clostridia in aDNA samples. These include the possibility of their introduction by human handling of archeological samples by researchers, their introduction from human handlers during ancient mummy preparation practices, or post-mortem colonization by environmental (e.g., soil) clostridia. Regarding our identification of *C. tetani* in numerous mummy samples, it is worth mentioning that although today mummies are manipulated taking in consideration cross-contamination, during the 1990s when some of these mummies were first discovered, fieldwork procedures were different. Thus, it is notable that the Chinchorro mummy sample from which we identified the TeNT/Chinchorro toxin, was not always handled by researchers using nitrile gloves during digging; also, these mummies have been manipulated by many researchers since their initial discovery (B. Arriaza, Personal communication).

Although we assessed all the samples for possible environmental microbial contamination based on environmental-specific microbial species, a more ideal scenario is if metagenomic sequencing data for local samples nearby the burial sites is also available for assessing sample contamination. For one of our identified acBins ("Tepos_35"), the authors of the original study had also sequenced DNA from soil samples near the burial site[64]. *C. tetani* DNA was detected not only in the Tepos_35 sample, but also in the nearby soil although at lower relative abundance[64]. Ultimately, this is consistent with the possibility that environmental *C. tetani* from soil may in some cases be the source of the *C. tetani* colonization of ancient human remains. If environmental or human contamination has resulted in the presence of *C. tetani* samples such as those described above, this explanation may account for the observation of low *C. tetani* damage rates in a portion of our dataset. For other samples where *C. tetani* damage levels are high (>10%) and thus indicative of an ancient origin, it remains unknown whether these strains are the result of ancient sample colonization by environmental strains, or whether they are as old as the archeological samples themselves.

Regardless of whether the identified *C. tetani* genomes are contemporaneous with the archeological samples, an important finding of this work is the substantial expansion of genomic knowledge surrounding *C. tetani* and its relatives, such as the expansion of clade 2 and clade 1H, as well as the discovery of *Clostridium* sp. X and Y. Lineage 1H in particular has undergone the greatest expansion through the newly identified aDNA-associated *C. tetani* genomes, from one known sample derived from a patient in France in 2016[10], to 9 additional draft genomes assembled from ancient DNA. This suggests that a broader diversity of 1H strains may exist in undersampled environments. Interestingly, these newly identified lineage 1H strains share a common pattern of originating from the Americas, which suggests that a lineage 1H *C. tetani* strain specific to (or abundant within) this region may have colonized these samples at some point in the past. acBins from lineage 2 also show some geographical clustering, again which could be due to the nature of *C. tetani* strains present within these regions and the time periods at which they colonized these samples. Some geographically clustered samples (e.g., Canary Island samples, and Egyptian samples) were actually associated with diverse lineages. Despite being close geographically, these samples have different dates of origin and estimated *C. tetani* damage levels. The diversity of *C. tetani* strains in these samples may reflect natural strain variation present within these regions, differences in strain composition over

time, and the possible influence of frequent travel to and from these regions throughout their history.

In addition to the expansion of existing lineages, our genomic analysis revealed two potentially new lineages of *Clostridium* that are closely related to, but distinct from, *C. tetani*. One of these (*Clostridium* sp. "Y") was assembled from an aDNA sample (GranCanaria-008-Tooth) taken from an archeological specimen dated to 936CE. The other potentially new lineage is *Clostridium* sp. "X", a group of closely related *Clostridium* strains that formed a sister lineage to *C. tetani* and yet are genomically distinct from *C. tetani* and all other available *Clostridium* genomes in the NCBI database. This clade is unlikely to have arisen by errors in genome sequencing or assembly artifacts as it is supported by the co-clustering of multiple genomes, its members are consistently placed as a divergent branch in marker gene phylogenies, its members show a consistent poor coverage alignment to the E88 reference chromosome, and it displays unique gene content and genomic characteristics that are conserved across multiple strains.

Although there appears to be some partial conservation of a *C. tetani*-like plasmid in *Clostridium* sp. X and Y with genes such as *repA* and *colT* consistently detected, the *tent* gene was either not detected or was fragmented, indicative of a pseudogenization event. These findings support the delineation of the species boundary between these groups, and highlight *tent* as a key gene separating *C. tetani* from closely related species. However, even if they are only pseudogenes, the detection of partial *tent* fragments in the X acBins raises the intriguing possibility that some members of *Clostridium* sp. X may have at one point encoded a full-length *tent* gene, and that *tent*-carrying plasmids may have been circulating in the past outside of *C. tetani* in other closely related *Clostridium* species. This is in line with recent genomic studies that have demonstrated the potential for species outside of the *Clostridium* genus to carry neurotoxin genes[57,65,66]. Future efforts to sequence the microbiomes associated with archeological samples, as well as environmental *Clostridium* isolates, may shed further light on these questions.

Beyond expanding *C. tetani* and *Clostridium* genomic diversity, our work also expands the known diversity of clostridial neurotoxins—the most potent family of toxins known to science. Analysis of ancient DNA revealed novel variants and lineages of TeNT, including the newly identified "subgroup 3" toxins: TeNT/Chinchorro, TeNT/El-Yaral toxins, and TeNT/Chiribaya-Alta. Not only do these toxins share a similar mutational profile, but they are derived from a similar geographic area (regions of Peru and Chile in South America) and their associated *C. tetani* genomes also cluster phylogenetically as the closest neighbors. Of the three subgroup 3 *tent* sequences identified, one of them (*tent*/Chinchorro) had sufficient coverage to be fully assembled, and the *tent*/Chinchorro gene also happened to be most divergent from modern *tent* sequences by possessing the greatest number of unique substitutions. Despite being the most divergent *tent*, reads mapping to the *tent*/Chinchorro gene as well as the associated acBin did not show strong patterns of DNA damage, and the damage level was weaker than that for human mtDNA. This indicates that, despite originating from the oldest sample in our dataset and possessing a unique *tent* variant, it is possible that the Chinchorro mummy-associated *C. tetani* strain may be a relatively "newer" strain that colonized the sample post-mortem.

Due to the uniqueness of TeNT/Chinchorro, and its collection of amino acid substitutions that have not been observed in any modern TeNT variants, we sought to determine whether this TeNT variant is a functional neurotoxin. Lack of toxicity, for instance, might indicate a sequence artifact or even a TeNT variant that targets other non-mammalian species. We therefore utilized a previous approach based on sortase-mediated ligation to produce small quantities of the full-length protein toxin[57,58], and compared the results to WT TeNT produced in the same manner. TeNT/Chinchorro produced a classic tetanus phenotype in mouse assays, and exhibits a potency

comparable to modern TeNT. This validated our predicted neurotoxic activity for this gene sequence, and suggests that TeNT/Chinchorro's multiple unique amino acid substitutions have a limited impact on potency and neurotoxicity. However, their non-random spatial clustering on a specific surface-exposed loop of the neurotoxin structure suggests that they may be a result of positive selection, a pattern that has been commonly observed in protein evolutionary studies[67,68]. Such substitutions may alter yet-to-be-identified TeNT protein-protein interactions. In addition to validating the predicted activity of TeNT/Chinchorro, this experimental method allowed us to directly test the disease-causing properties of a novel virulence factor variant reconstructed from aDNA, without the necessity for growing the organism itself, which would have biosafety concerns.

In summary, using large-scale data mining, we identified ancient neurotoxigenic clostridia in archeological samples. This resulted in a substantial expansion of the known genomic diversity and occurrence of *C. tetani* and led to the discovery of novel *C. tetani* lineages, potentially new *Clostridium* lineages, and tetanus neurotoxin variants that retain functional activity. The discovery of neurotoxigenic clostridial genomes in such a wide diversity of ancient samples, both geographically and temporally, is unexpected, but perhaps not inconsistent with prior hypotheses about the role of these organisms and other toxigenic pathogens in the natural decomposition process[66,69,70]. More generally, *Clostridium*-derived DNA is common in ancient DNA collected from human tissues including but not limited to paleofecal microbiomes[17] as well as in microbial communities associated with vertebrate decomposition[71,72].

Despite our identification of neurotoxigenic *C. tetani* and related species in aDNA samples, there are several limitations of our study. These include the limitation of only analyzing the top candidate samples identified from the NCBI SRA and issues with taxonomic binning and genome assembly of ancient metagenomes. In particular, the main technical limitation of our study is our inability to determine the extent that the identified *C. tetani* and related genomes are derived from one or multiple strains. This leads to some uncertainty in the phylogenetic trees (and estimated damage levels) that we calculated because the patterns we observed may result from a mixture of signals coming from different strains. Lastly, although the damage levels of the identified *C. tetani* and related *Clostridium* genomes are indicative of ancient DNA, the precise origin of these strains remains unknown for the reasons described earlier. Despite this, we anticipate that further exploration of ancient archeological samples will shed further light on the genomic and functional diversity of these fascinating organisms, as well as the ecology and evolutionary origins of their remarkably potent neurotoxins.

## Methods

### Identification of *C. tetani* containing samples from the NCBI sequence read archive

To identify datasets within the NCBI sequence read archive predicted to contain *C. tetani* DNA, we searched the NCBI-stat database[30] on March 15, 2021 for matches to tax_id = 1513 using Google's Big Query API (Cloud SDK v330.0.0). The query returned 43,620 sample hits with a *k*-mer self-count ranging from 1 to 17,152,980. A threshold of 23,000 was applied which returned 136 hits including 28 *C. tetani* sequencing projects (which were used as controls). The STAT taxonomic profile was retrieved using a separate Google Big Query search and was used to assess the microbial community in each sample. Total counts of each mapped bacterial and archaeal taxon at the species level were extracted from the profile and were converted to proportional values and subsequently visualized in R v4.0.4.

FASTQ files of identified sequencing runs were downloaded to Digital Research Alliance of Canada (formerly Compute Canada) infrastructure using the sra-toolkit v2.9.6, and quality encodings of all runs were assessed. Eight runs were Phred+64 encoded and were

converted to Phred+33 using seqtk v1.3[73]. Twenty-three runs had an unknown encoding and were assumed to be Phred+33 encoded based on the range of the quality scores. The remaining runs were Phred+33 encoded.

### Recovery and QC assessment of *C. tetani*-related genomic sequences from ancient DNA samples

Reads were pre-processed using fastp v0.20.1[74] with default settings to perform quality filtering and remove potential adapters. Pre- and post-processing statistics are included in Supplementary Data 22. Metagenome co-assembly, using all reads with the same BioSample ID, was performed using MEGAHIT v1.2.9 with default parameters[39]. Contigs were then taxonomically classified using Kaiju v1.7.4[40] against the Kaiju database nr 2021-02-24 with default settings. Any contigs that mapped to *C. tetani* (NCBI taxonomy ID 1513) or any of its strains (NCBI taxonomy IDs 1231072, 212717, 1172202, and 1172203) were selected for further analyses. The kaiju-identified *C. tetani* contigs were further screened via a BLASTN search with Blast+ v2.12.0 against all *Clostridium* genomes from Refseq on Aug. 25, 2022. Any contigs with a *C. tetani* top hit were labeled as *C. tetani* BLAST-validated. Contigs containing rRNA, tmRNA, and tRNA were also removed from acBins prior to phylogenetic analysis, as these sequences were associated with high-coverage outlier regions due to increased mapping of related reads from non *C. tetani* species. Barrnap v0.9 [https://github.com/tseemann/barrnap] was used with (--reject 0.05) to sensitively identify rRNA sequences in the metagenomic contigs, including hits up to and including 5% of the expected length. Aragorn v1.2.36[75] was used to identify tRNA and tmRNA sequences with -ps to slightly lower scoring thresholds (to 95% of default).

For QC analysis, CheckM v1.0.18[41] was used to assess acBins for completeness, contamination, and strain heterogeneity, with the pre-built set of *Clostridium* markers supplied with the tool. To examine strain variation, we used a custom Python script to analyze the mpileup results from samtools v1.15.1[76] to quantify the mean per-base heterogeneity for each acBin. This involved calculating the percentage of bases for a position in disagreement with the reference base and calculating the overall mean across all non-zero coverage positions.

ANI calculations were performed using two approaches: one method used fastANI v1.33[54] to compare the contig sets for each acBin with *C. tetani* and other *Clostridium* genomes; the second method calculated ANI directly from the snippy-generated multiple genome alignment using the dist.alignment() function from the seqinr v4.2-16 R package[77]. Scripts associated with the above analyses and contig sets and for three stages of the analysis pipeline (Kaiju, Kaiju+BLAST, Kaiju+BLAST + RNA-removed) are available in the public github repository.

### Measurement and visualization of genome coverage and alignments

Genome coverage for the recovered acBins was further assessed and visualized in two ways. First, Bowtie2 v2.4.2[78] was used to map reads from individual runs to the E88 chromosome (accession NC_004557.1) and plasmid (accession NC_004565.1). Bowtie2 was run with the following parameters (--local -D 20 -R 3 -N 1 -L 20 -i S,1,0.50 -bSF4). Using samtools v1.12, the resultant BAM files were then sorted, indexed, and merged based on their BioSample ID. Total (average # of reads per base) and percent (number of bases with 1 or more reads divided by total number of bases) coverage was calculated for the entire chromosome and plasmid as well as the *tent*, *colT*, and *repA* regions. Coverage was visualized using Python v3.8.5 and matplotlib v3.3.2. Coverage plots (Supplementary Fig. 17) were created by loading BAM files into R v4.1.0 with the Rsamtools library v2.8.0 and plotted as area plots using ggplot2 v3.3.5. Coverage of the plasmid

sequences was calculated by averaging the number of reads per base in 100-bp bins.

As a second approach, we also visualized the multiple genome alignment generated by snippy (described below) using ggplot2 v3.3.5 after loading it into R v4.1.0 using Biostrings 2.62.0.

### Analysis of ancient DNA damage

Fastq files were pre-processed using leeHom v1.2.15[79] to remove adapters and to perform Bayesian reconstruction of aDNA with the (--ancientdna) flag applied to paired-end datasets. The leeHom output was then merged by bioSample ID (concatenated sequentially into one file). Individual and merged results were then processed using seqtk v1.3 with the "seqtk seq -L30" command to remove short sequences <30 bp in length. For each BioSample, trimmed reads were then mapped using BWA v0.7.17-r1188[80] to the contigs that classified as *C. tetani* using Kaiju[40] and BLAST with rRNAs, tRNAs, and tmRNAs removed as described above. Trimmed reads were separately mapped to the human mitochondrial reference genome (accession NC_012920.1). Read alignment was performed using "bwa aln" with the (-n 0.01 -o 2 -l 16500) options. BAM files were sorted using samtools v1.12. Misincorporation rates were then measured in two ways: (1) for all samples using mapDamage v2.2.1[42] with (--merge-reference-sequences --no-stats); and (2) using pyDamage v0.70[43] to estimate DNA damage on a per contig basis with default parameters and per assembly basis with the (--group) parameter.

### Whole genome alignments and phylogenetic reconstruction

All available *C. tetani* genomes ($N = 43$) were downloaded from the NCBI Genbank database. After duplicate strains were removed, these included the 37 genomes from ref. 10 as well as four additional genomes that were not included in ref. 10 but were present in the NCBI (Supplementary Data 23). To compare our 38 acBins with the 41 non-redundant *C. tetani* genomes and investigate their phylogenetic relationships, we used two approaches described below.

Single-base substitutions within assembled *C. tetani* contigs were identified using snippy-multi from the Snippy package v4.6.0 [https://github.com/tseemann/snippy] with *C. tetani* str. E88 as the reference genome (GCA_000007625.1_ASM762v1_genomic.gbff). A genome-wide core alignment was constructed using snippy-core. Five aDNA samples (Deir-Rifeh-KNIII-Tooth, Deir-Rifeh-KNII-Tooth, Pericues-BC28-Bone, Tepos_35-Tooth, Barcelona-3031-Tooth) were removed due to very poor alignment coverage (<1%). Using the resulting alignment, we built a phylogeny using FastTree v2.1.10[81] with the GTR model and aLRT metric for assessment of clade support. A maximum-likelihood phylogeny was also constructed using RAxML v8.2.12[82] with a GTR + GAMMA substitution model and 1000 rapid bootstrap inferences. The alignment and phylogeny were analyzed for potential recombination with Gubbins v3.3[83] and visualized with Phandango v.1.3.0[84]. The whole genome alignment and trees can be found in the public github repository.

We also used Harvest suite v1.2, which provides an automated pipeline for generating and visualizing core-genome alignments, SNP detection, and phylogenetic analysis of intraspecific microbial strains[47]. Parsnp v1.2 with the recombination filtration option (-x) was used to examine all acBins and modern *C. tetani* genomes and produce core alignments of orthologous sequences. Since Parsnp only identifies core blocks shared across a group of highly similar genomes, it effectively filters out low-coverage and divergent genomes. All 41 modern strains but only 11 acBin strains with MUMi distance values <= 0.20 were kept in the final Parsnp core alignment. FastTree[81], as included within the Harvest pipeline, was used to produce the final phylogeny. Gingr v1.3 was used to visualize the alignment variation pattern and tree. The resulting tree can be found in the public github repository.

### Annotation and comparative genomics of acBins with modern *C. tetani* strains

We annotated all contigs from the 38 acBins strains using prodigal v2.6.2[85]. Anonymous annotation mode was used as it is intended for metagenomes and low-quality draft genomes. Sequences encoding rRNA, tRNA and tmRNAs were also annotated as described above. These annotations resulted in predicted proteomes for all 38 acBins, which were then compared with the existing proteomes from the 41 modern *C. tetani* genomes. To cluster identified protein-coding sequences into groups of orthologous sequence clusters, we used Orthofinder v2.5.4[86] with default settings. Orthofinder was applied to modern *Clostridium tetani* proteomes and prodigal-predicted acBin proteomes, resulting in a set of 4334 total orthogroups. Any orthogroup that was made up of only partial coding sequences (as predicted by prodigal) was removed to limit orthogroups resulting from gene fragments or pseudogenes. Orthogroups without any coding sequences found on *C. tetani* BLAST-validated contigs were also removed as a filter for any potential bin contamination. We then performed a gene set comparison between acBins and modern strains by comparing the presence/absence of all identified orthogroups. This resulted in the identification of common orthogroups shared between modern and acBin strains as well as orthogroups unique to acBins, which were investigated further through gene neighborhood analysis. In particular, we focused on orthogroups unique to *Clostridium* sp. X and investigated their gene neighborhoods and their conservation with modern reference *C. tetani* genomes. For gene neighborhood visualization and alignment, we used AnnoView v1.0 (annoview.uwaterloo.ca).

### Sequence, structural, and phylogenetic analysis of ancient tetanus neurotoxins

**Variant calling and construction of the MSA.** Scripts used for variant calling and generation of a *tent* multiple alignment are located in the public github repository. For the plasmid read alignments used earlier, we extracted aligned reads, and re-aligned them using BWA (bwa-mem) v0.7.17-r1188 using default parameters. Read alignments were manipulated with samtools v1.12 and htslib v1.12. The read alignment was restricted to the *tent* gene locus for variant calling (using the reverse complement of NC_004565.1, bases 1496-5443). Variants were called on each individual sample using the Octopus variant caller v0.7.4[87] with stringent parameters (--mask-low-quality-tails 5 --min-mapping-quality 10 --min-variant posterior 0.95 --min-pileup-base-quality 35 --min-good-base-fraction 0.75). This combination of parameters reports only variants with very high confidence and read mapping quality, minimizing the identification of false positive variant calls. We then built consensus sequences of *tent* genes from each sample using the bcftools consensus tool v1.12, and htslib v1.12, replacing positions with 0 coverage with a gap character. MAFFT v7.4.80[88] was used to realign fragments against the reference sequence using the (--keeplength) option, which notably keeps the length of the reference unchanged and therefore ignores the possibility of unique insertions. The final *tent* alignments are available in the public github repository.

**Structural modeling.** A structural model of TeNT/Chinchorro was generated by automated homology modeling using the SWISSMODEL server[89]. Modeling was performed using two top-scoring homologous template structures of tetanus neurotoxins: PDB IDs 7by5.1.A [https://www.rcsb.org/structure/7by5] (97.18% identity), 5N0C.1.A [https://www.rcsb.org/structure/5N0C] (97.34% identity). 7BY5.1.A was selected as the best template based on the QMEAN quality estimate[90]. The model was visualized using PyMOL v2.4.1 and unique substitutions (present in TeNT/Chinchorro but absent in modern TeNT sequences) were highlighted.

**Phylogenetic analysis.** The *tent* consensus alignment generated as described earlier was processed to keep only sequences ($N = 20$) with alignment coverage exceeding 80%. The following BioSamples were removed: Tepos_35-Tooth, Tenerife-012-Tooth, Tenerife-011-Tooth, SLC-France-Tooth, Peru-NA42-Bone, Deir-Rifeh-KNII-Tooth, Peru-NA40-Bone, Punta_Candelero-Tooth, Gniezno-Tooth, Augsburg-Tooth, Riga_Cemetery-Tooth, Pericues-BC28-Bone, Marseille-Tooth. We then aligned the 20 ancient *tent* gene sequences with 30 *tent* sequences from modern *C. tetani* strains, which reduced to 12 representative modern *tent* sequences after duplicates were removed using Jalview v2.9.0b2. *tent*/E88 was identical with *tent* from 11 strains (1586-U1, CN655, 641.84, C2, Strain_3, 75.97, 89.12, 46.1.08, A, 4784 A, Harvard), *tent*/132CV with 1 other (Mfbjulcb2), *tent*/63.05 with 2 others (3483, 184.08), *tent*/1337 with 2 others (B4, 1240), *tent*/ATCC_453 with 1 other (3582), and *tent*/202.15 with 1 other (358.99). A phylogeny was constructed using PhyML v3.1[91] with the GTR model, empirical nucleotide equilibrium frequencies, no invariable sites, across site rate variation optimized, NNI tree search, and BioNJ as the starting tree. PhyML analysis identified 362 patterns, and aLRT (SH-like) branch supports were calculated. The final newick tree is available in the public GitHub repository.

**Experimental testing of TeNT/Chinchorro (chTeNT)**
**Antibodies and constructs.** Antibodies for Syntaxin-1 (HPC-1), SNAP25 (C171.2), VAMP1/2/3 (104102) were purchased from Synaptic Systems. Antibody against actin (AC-15) was purchased from Sigma. Rabbit antiserum of TeNT (ab53829) was purchased from Abcam, rabbit non-immunized serum (AB110) was purchased from Boston Molecules. The cDNAs encoding chTeNT-LC-$H_N$ (the N-terminal fragment, residues 1–870) and chTeNT-$H_C$ (the C-terminal fragment, residues 875–1315) were synthesized by Twist Bioscience (South San Francisco, CA). The cDNA encoding TeNT-LC-$H_N$ (residues 1–870) and TeNT-$H_C$ were synthesized by GenScript (Piscataway, NJ). The full WT TeNT sequence can be found at Uniprot (accession # P04958). A thrombin protease cleavage site was inserted between I448 and A457 in both TeNT-LC-$H_N$ and chTeNT-LC-$H_N$. LC-$H_N$ fragments were cloned into pET28a vector, with peptide sequence LPETGG fused to their C-termini, followed by a His6-tag. $H_C$ fragments were cloned into pET28a vectors with a His6-tag and thrombin recognition site on their N-termini.

**Protein purification.** *E. coli* BL21 (DE3) was utilized for protein expression. In general, transformed bacteria were cultured in LB medium using an orbital shaker at 37 °C until the $OD_{600}$ reached 0.6. Induction of protein expression was carried out with 0.1 mM IPTG at 18 °C overnight. Bacterial pellets were collected by centrifugation at $4000 \times g$ for 10 min and disrupted by sonication in lysis buffer (50 mM Tris pH 7.5, 250 mM NaCl, 1 mM PMSF, 0.4 mM lysozyme), and supernatants were collected after centrifugation at $20,000 \times g$ for 30 min at 4 °C. Protein purification was carried out using a gravity nickel column, then purified proteins were desalted with PD-10 columns (GE, 17-0851-01) and concentrated using Centrifugal Filter Units (EMD Millipore, UFC803008).

**Sortase ligation.** $H_C$ protein fragments were cleaved by thrombin (40 mU/µL) (EMD Millipore, 605157-1KU) overnight at 4 °C. Ligation reaction was set up in 100 µL TBS buffer with LC-$H_N$ (8 µM), $H_C$ (5 µM), $Ca^{2+}$ (10 mM) and sortase (1.5 µM), for 1 h at room temperature. Full-length proteins were then activated by thrombin (40 mU/µL) at room temperature for 1 h. Sortase ligation reaction mixtures were analyzed by Coomassie blue staining and quantified by BSA reference standards.

**Neuron culture and immunoblot analysis.** Primary rat cortical neurons were prepared from E18-19 embryos using a papain dissociation kit (Worthington Biochemical) following the manufacturer's instruction. Neurons were exposed to sortase ligation mixtures with or without antiserum in culture medium for 12 h. Cells were then lysed with RIPA buffer with a protease inhibitor cocktail (Sigma-Aldrich).

Lysates were centrifuged at $12,000 \times g$ at 4 °C for 10 min. Supernatants were subjected to SDS–PAGE and immunoblot analysis.

**Animal study.** All animal studies were approved by the Boston Children's Hospital Institutional Animal Care and Use Committee (Protocol Number: 18-10-3794 R). Toxins were diluted using phosphate buffer (pH 6.3) containing 0.2% gelatin. Mice (CD-1 strain, female, purchased from Envigo, 6–7 weeks old, 25–28 g, $n = 3$ per group, 9 total) were anesthetized with isoflurane (3–4%) and injected with toxin (10 µL) using a 30-gauge needle attached to a sterile Hamilton syringe, into the gastrocnemius muscles of the right hind limb with the left leg serving as the negative control. Muscle paralysis was observed for 4 days. The severity of spastic paralysis was scored with a numerical scale modified from a previous report (0, no symptoms; 4, injected limb and toes are fully rigid)[92].

## Reporting summary
Further information on research design is available in the Nature Portfolio Reporting Summary linked to this article.

## Data availability
Raw genomic data reported in this study is available from the NCBI sequence read archive. Accession numbers for all BioSamples and sequencing runs used are listed in Supplementary Data 1 and 2. Processed data generated in this study are provided in the Supplementary Data and Source Data files. Additional large datasets generated by this study have been deposited in FigShare under the following accession numbers: https://doi.org/10.6084/m9.figshare.21498198, https://doi.org/10.6084/m9.figshare.21498222, https://doi.org/10.6084/m9.figshare.21498330, https://doi.org/10.6084/m9.figshare.21652340, and https://doi.org/10.6084/m9.figshare.23804106. Source data are provided with this paper.

## Code availability
Source code for this project and additional links to datasets are available for download by Hodgins et al.[93], and can also be accessed at https://github.com/harohodg/aDNA-tetanus-analysis.

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

## Acknowledgements

This study was supported by the Natural Sciences and Engineering Research Council (NSERC) through a Discovery Grant (RGPIN-2019-04266) and Discovery Accelerator Supplement (RGPAS-2019-00004) awarded to A.C.D., by the Government of Ontario through an Early Researcher Award to A.C.D, and by a University of Waterloo Interdisciplinary Trailblazer grant awarded to A.C.D. and A.E.D. A.C.D. also holds a University Research Chair from the University of Waterloo. M.J.M. gratefully acknowledges funding from the Japan Society for the Promotion of Science as a JSPS International Research Fellow (Luscombe Unit, Okinawa Institute of Science and Technology Graduate University). H.P.H. gratefully acknowledges funding from an NSERC Canada Graduate Scholarship. H.P.H. and A.C.D. also acknowledge the Digital Research Alliance of Canada (formerly Compute Canada) for providing access to high-performance computing resources. This study was also partially supported by grants from National Institute of Health (NIH) (R01NS080833 and R01NS117626 to M.D). M.D. holds the Investigator in the Pathogenesis of Infectious Disease award from the Burroughs Wellcome Fund.

## Author contributions

A.C.D. conceived and supervised the project. H.P.H., B.T., B.L., M.J.M, V.L., X.W., and A.C.D. performed bioinformatic data analysis. A.C.D., G.R., and A.T.D. supervised aDNA analysis. P.C. and P.L. performed all experimental work, which was supervised by M.D., A.E.D., and J.C. performed context analysis of archeological samples. All authors contributed to the manuscript writing and preparation of figures.

## Competing interests

The authors declare no competing interests.
