## [Peer Review File - Redacted NEW · Nature Communications]

Ancient Clostridium DNA and variants of tetanus neurotoxins associated with human archaeological remainsEditorial Note: This manuscript has been previously reviewed at another journal that is not operating a transparent peer review scheme. This document only contains reviewer comments and rebuttal letters for versions considered at *Nature Communications*. Mentions of the other journal have been redacted.

Reviewer #1 (Remarks to the Author):

The present paper largely expands the present knowledge on pathogens associated to archeological human samples with a unique and study dedicated to a bacterial toxin that by itself has been, and still is in large parts of the world, a major killer of humans and other vertebrates. The genomics of *C. tetani* is revolutionized and expanded by the present work.

The work was submitted to another journal [Redacted]. The present paper reads much better. Errors present in the previous paper have been removed and almost all the suggestions made by the Referees were appropriately taken into consideration and elaborated in the new version of the work.

The unavoidable limit of the work related to the fact that the archaeological material may have been contaminated in an indeterminable time following death has been discussed in different parts of the text. Likewise, some ingenuities present in the first text were removed.

The work is as complete as possible regarding to the characterization of tetanus toxin sequences and activity of an "antique" toxin also thanks to the addition of supplementary material . In addition clades of *C. tetani* from archaeological reports derived from different parts of the world were identified and characterized. This largely expands the present knowledge on pathogens associated to archeological human samples with a unique and first study dedicated to a bacterial toxin that by itself has been, and still is in large parts of the world a major killer of humans and other vertebrates.

Reviewer #2 (Remarks to the Author):

In this revised version, the authors have expanded their list of collaborators to include at least two additional coauthors who have experience in ancient DNA/biological anthropology (AD and AD). While the authors have made some relevant revisions to their discussion regarding the challenges with authenticity of these data, problematic passages persist in the manuscript. Obtaining reads from an environmental organism from an ancient DNA dataset that is likely dominated by environmental microbial content does not mean that the sequences are contemporaneous with the archaeological tissue. Highly problematic passages in my opinion can be found in the following lines of text: 42-44, 48 – 50, 90-109 (this passage deserves a transparent discussion of the retrievability of ancient molecules, and the fact that nearly all ancient DNA extracts are dominated by non-endogenous content... this also makes the content of line 138 completely expected), 111-112, and 152-154 (higher quality references with quantitative data would offer better support for the statement).

Line 161 – 162: are these values based on percentages of identifiable reads, or total bulk DNA content? The majority of reads in an ancient dataset do not receive a taxonomic assignment.

Line 162: (and 179 and Figure S4) – with what metric was contamination defined?

Line 180 (and Figure S3): Have the authors normalised these values over sequencing depth? Capture dataset are sometimes sequenced to greater depth than shotgun attempts (though not always), and I would like to see this statement made in such a way to consider number of reads in the dataset. I would also be rather surprised if enrichment, at least for another pathogen, reduced recoverable *C. tetani* reads, as the proportion of target pathogen in a typical dataset is low even after two rounds of enrichment. The situation may well be different for human genomic captures. I'm curious if the authors simply mapped their datasets to a *C. tetani* reference to determine the % *C. tetani* DNA in the various datasets. That might be clearer and easier to interpret than the assembly sizes and completeness plots in figure S3.

Prior to presentation of the phylogenetic tree, the main manuscript should include a table that discloses basic genomic reconstruction metrics such as sample name, age of the tissue, average genomic coverage of *C. tetani*, % of the *C. tetani* genome covered at X-fold (assuming they do

SNP calling via an established method that considers coverage), 5' damage in *C. tetani*, and 5' damage in human mtDNA.

Figure S5 – the authors should specify that the datasets are shown in decreasing 5' damage from Clostridia mapping

Figure S7 - the caption does not match the plots shown in a - d. Also use of the same colors in c as those in a and b is confusing. Midpoint date axis should be in YBP. Also, it seems rather strange that, for regular inhumation, damage increases with younger datasets. That is not what your damage plots in Fig S5 show, where the two libraries with the highest number of damaged reads (Augsburg and SLC) do not correspond temporally with the highest damage values in the plot. Something seems incorrect with the midpoint calculation

Figure S8 - there is no legend to explain the color scheme. How is it possible to show consistent colors for "Total SNPs", "SNPs inside recombination" and "SNPs outside recombination"? Unless I have misunderstood, this would imply some kind of Schroedinger-like system wherein all positions are simultaneously recombinant and non-recombinant. In panel A the dendrogram is too collapsed to make any sense of it. What are the units in the line plot below that range from what looks to be 0 to 36?

Figure S9 - Rather than calling these "high coverage acBins", I suggest the authors define their threshold for inclusion, as "high coverage" is a subjective statement. I assume the depth of the purple color corresponds to the density of contigs that map along the reference, and I assume your reference is from lineage 2, hence the reason for higher density of the lineage 2 bins? Importantly the misincorporation plots of Sanganji-A1-Tooth show a lot of grey, which means there is high non-target background in the mapping reads. The same is shown for Sanganji-A2-Tooth (3' end). Tenerife 004 and 013, and the Chincurro mummy, have discrepant damage plots between human mt and acBins which questions their authenticity. Also, Abusir1595-Tooth seems to have no human DNA from the misincorporation plot. Is this figure based on bins post-recombination filtering? That is not stated. Of note, multiple genomes included here were identified as problematic based on heterozygous proportion in figure S4. Prior to their further analysis, a plot similar to S4 should be shown to demonstrate the effect of applying the Gubbins/Phandango filters.

Based on the phylogeny shown in Figure 1c that demonstrates the derived positions of several of their putative ancient acBins, I remain unconvinced that several of these genomes derive from an ancient context.

Reviewer #4 (Remarks to the Author):

The authors have been exceedingly thorough in their responses to my comments, including appropriate modifications to the text and in some instances including updated information. My initial overall evaluation was favorable, and I have no additional concerns with the revised manuscript.

REVIEWER COMMENTS

Reviewer #1 (Remarks to the Author):

The present paper largely expands the present knowledge on pathogens associated to archeological human samples with a unique and study dedicated to a bacterial toxin that by itself has been, and still is in large parts of the world, a major killer of humans and other vertebrates. The genomics of *C. tetani* is revolutionized and expanded by the present work.

The work was submitted to another journal [Redacted]. The present paper reads much better.

Errors present in the previous paper have been removed and almost all the suggestions made by the Referees were appropriately taken into consideration and elaborated in the new version of the work.

The unavoidable limit of the work related to the fact that the archaeological material may have been contaminated in an indeterminable time following death has been discussed in different parts of the text. Likewise, some ingenuities present in the first text were removed.

The work is as complete as possible regarding to the characterization of tetanus toxin sequences and activity of an "antique" toxin also thanks to the addition of supplementary material . In addition clades of *C. tetani* from archaeological reports derived from different parts of the world were identified and characterized. This largely expands the present knowledge on pathogens associated to archeological human samples with a unique and first study dedicated to a bacterial toxin that by itself has been, and still is in large parts of the world a major killer of humans and other vertebrates.

We thank the reviewer for their positive comments.

Reviewer #2 (Remarks to the Author):

In this revised version, the authors have expanded their list of collaborators to include at least two additional coauthors who have experience in ancient DNA/biological anthropology (AD and AD). While the authors have made some relevant revisions to their discussion regarding the challenges with authenticity of these data, problematic passages persist in the manuscript. Obtaining reads from an environmental organism from an ancient DNA dataset that is likely dominated by environmental microbial content does not mean that the sequences are contemporaneous with the archaeological tissue. Highly problematic passages in my opinion can be found in the following lines of text: 42-44, 48 – 50, 90-109 (this passage deserves a transparent discussion of the retrievability of ancient molecules, and the fact that nearly all ancient DNA extracts are dominated by non-endogenous content... this also makes the content of line 138 completely expected), 111-112, and 152-154 (higher quality references with quantitative data would offer better support for the statement).

We have reworded sentences on lines 42-44 and 48-50 to remove mention of the age of the samples.

Lines 90-109 (95-104 in current version): We have added a discussion on the issues regarding contamination by non-endogenous content and cited three relevant papers.

111-112 (119-120 in current version): We reworded this to clarify that our newly identified strains and neurotoxin genes are from “**datasets associated with human archaeological samples**”, in order to remove the implication that the DNA is all ancient.

152-154 (155-158 in current version): We added higher quality references and quantitative data here describing the % microbial DNA content of the samples.

Line 161 – 162: are these values based on percentages of identifiable reads, or total bulk DNA content? The majority of reads in an ancient dataset do not receive a taxonomic assignment.

We have clarified this on what is now line 165. Here, we were interested in examining the percentage of environment-specific microbes within the “**microbially-classified reads**”.

Line 162: (and 179 and Figure S4) – with what metric was contamination defined?

We have added to the text what metrics were used in all three of these cases.

Line 162 (now 166) - “proportions of reads from putative environment-specific microbes”

Line 179 (now 183) - “checkM contamination”

Figure S5 - also “checkM contamination”

Line 180 (and Figure S3): Have the authors normalised these values over sequencing depth? Capture dataset are sometimes sequenced to greater depth than shotgun attempts (though not always), and I would like to see this statement made in such a way to consider number of reads in the dataset. I would also be rather surprised if enrichment, at least for another pathogen, reduced recoverable *C. tetani* reads, as the proportion of target pathogen in a typical dataset is low even after two rounds of enrichment. The situation may well be different for human genomic captures. I'm curious if the authors simply mapped their datasets to a *C. tetani* reference to determine the % *C. tetani* DNA in the various datasets. That might be clearer and easier to interpret than the assembly sizes and completeness plots in figure S3.

As suggested by the reviewer, we now present this information in two additional ways that normalize to sequencing depth (see newly added Figure S3C and D). We calculated the % *C. tetani* DNA from the initial STAT taxonomic profiling, and calculated the average depth of coverage of the *C. tetani* chromosome normalized by dataset size.

Normalizing over sequencing depth reduced the observed difference, but it is still true that the shotgun datasets are associated with a greater amount of *C. tetani* DNA content.

We have reworded what is now lines 184-186 to reflect this change, removing our previous statement that this pattern could be due to capture-related enrichment of other organisms. We agree with the reviewer that this is not necessarily the cause of this pattern.

Prior to presentation of the phylogenetic tree, the main manuscript should include a table that discloses basic genomic reconstruction metrics such as sample name, age of the tissue, average genomic coverage of *C. tetani*, % of the *C. tetani* genome covered at X-fold (assuming they do SNP calling via an established method that considers coverage), 5' damage in *C. tetani*, and 5' damage in human mtDNA.

All of this information is actually already in our Supplementary Data Tables. Sample name, age of tissue, and 5' damage rates (*C. tetani* and mtDNA) are in Table S2, which was intended to be our "summary table" including key metrics as the reviewer mentioned. The average genomic coverage of *C. tetani* and % of *C. tetani* covered at 1-fold up to 10-fold is in Table S17.

Figure S5 – the authors should specify that the datasets are shown in decreasing 5' damage from Clostridia mapping

This has been done (see last sentence in Figure S5 caption).

Figure S7 - the caption does not match the plots shown in a - d. Also use of the same colors in c as those in a and b is confusing. Midpoint date axis should be in YBP. Also, it seems rather strange that, for regular inhumation, damage increases with younger datasets. That is not what your damage plots in Fig S5 show, where the two libraries with the highest number of damaged reads (Augsburg and SLC) do not correspond temporally with the highest damage values in the plot. Something seems incorrect with the midpoint calculation

We thank the reviewer for catching this error. The letter labels were mixed up in the caption. This has now been corrected. We have also changed the colors in C to distinguish it better from the others.

The Augsburg and SLC samples are correct in Fig S7c (top right data points in plot, highest damage but recent in date). This pattern may reflect that damage rates can be influenced by other factors than the age of the sample. The reviewer's confusion about the midpoint date may stem from the fact that we are showing the date in BCE and CE format, not YBP. We have updated the axis of the plot in Fig S7c to reflect this.

Figure S8 - there is no legend to explain the color scheme. How is it possible to show consistent colors for “Total SNPs”, “SNPs inside recombination” and “SNPs outside recombination”? Unless I have misunderstood, this would imply some kind of Schrodinger-like system wherein all positions are simultaneously recombinant and non-recombinant. In panel A the dendrogram is too collapsed to make any sense of it. What are the units in the line plot below that range from what looks to be 0 to 36?

We agree that Figure S8a (auto-generated image from the Phandango software tool) is too busy and collapsed to make sense of this data. S8A is actually not informative for the manuscript (it was never even referenced specifically in text) and the relevant information is what was presented in previous Figure S8b. We have therefore removed S8a and kept S8b.

Figure S9 - Rather than calling these “high coverage acBins”, I suggest the authors define their threshold for inclusion, as “high coverage” is a subjective statement. I assume the depth of the purple color corresponds to the density of contigs that map along the reference, and I assume your reference is from lineage 2, hence the reason for higher density of the lineage 2 bins?

That is a good point, and so we have removed the term “high coverage acBins”. To clarify, these acBins are the ones that passed Parsnp’s thresholds (i.e., MUMi distance values ≤ 0.20 and other default Parsnp thresholds). This is now described in the Methods section (line 699) and we also updated the text on lines 258-259.

The purple actually reflects the SNP density. Our reference is within clade 1A. We have updated the figure legend to describe this in more detail.

Importantly the misincorporation plots of Sanganji-A1-Tooth show a lot of grey, which means there is high non-target background in the mapping reads. The same is shown for Sanganji-A2-Tooth (3’ end).

Inspection of these two datasets revealed that: for an unknown reason, even after read trimming, the edges of a small proportion of reads have short non-adaptor sequence fragments that do not map correctly to the reference, whereas the rest of the sequences are clearly of *C. tetani* origin (based on BLAST alignments to the full nr database). We have made a note in the Figure S5 legend that these two samples were flagged as having problematic alignments near the edges of reads.

Tenerife 004 and 013, and the Chincurro mummy, have discrepant damage plots between human mt and acBins which questions their authenticity.

We agree that there are discrepant damage plots, and have further clarified this point on lines 217-220. Lower damage rates observed in the acBins may suggest clostridial colonization of the archaeological samples at a later date.

Also, Abusir1595-Tooth seems to have no human DNA from the misincorporation plot. Is this figure based on bins post-recombination filtering? That is not stated.

We thank the reviewer for pointing this out to us. The lack of human DNA in Abusir1595 is consistent with the STAT taxonomic profiling of reads from this dataset (see Figure S1). In fact, there are six datasets with an extremely low (<50) number of reads that mapped to human mtDNA: Deir-Rifeh-KNII-Tooth, Abusir1655-Tooth, Abusir1671-Tooth, Abusir1607-Tooth, Abusir1595-Tooth, and Vac-Mummy-Tissue.

We have therefore removed the mtDNA plots for these samples from Figure S5, updated the S5 legend, and also removed these data points from Figure S6 (the R^2 value was virtually unchanged - from 0.46 to 0.45).

Of note, multiple genomes included here were identified as problematic based on heterozygous proportion in figure S4. Prior to their further analysis, a plot similar to S4 should be shown to demonstrate the effect of applying the Gubbins/Phandango filters.

This is a good suggestion. We re-analyzed the percent base heterogeneity of mapped reads for the higher-quality subset of acBins that were included in the final Parsnp alignment in Figure 1c. We only assessed reads that mapped to contigs associated with this recombination-filtered alignment, that mapped with quality ≥ 30 and were not in high-coverage (>3 SD from mean depth of coverage) outlier regions.

As suspected by the reviewer, these filters improved the heterogeneity values. We have added a new plot (Figure S4b) and a line in the text referencing this new result (line 261).

Based on the phylogeny shown in Figure 1c that demonstrates the derived positions of several of their putative ancient acBins, I remain unconvinced that several of these genomes derive from an ancient context.

Since the initial version of our manuscript, we have added discussion regarding the putative origins of the clostridial DNA in the archaeological samples including post-mortem colonization by environmental strains. Although a subset of the acBins show misincorporation profiles indicative of ancient DNA, this does not necessarily suggest that these strains are contemporaneous with the archaeological samples, and we therefore do not make such a claim in our manuscript.

We want to emphasize the point that, even if the clostridial genomes are not as ancient as the samples themselves and originated later, we feel that the discovery of new *C. tetani* strains, related species (e.g., X), and novel toxin variants, is of significant interest to the microbiology community.

Reviewer #4 (Remarks to the Author):

The authors have been exceedingly thorough in their responses to my comments, including appropriate modifications to the text and in some instances including updated information. My initial overall evaluation was favorable, and I have no additional concerns with the revised manuscript.

We thank the reviewer for their positive comments.

Reviewer #2 (Remarks to the Author):

I thank the authors for providing a revised version of their manuscript. I have a few additional points to raise.

Line 43: "We assembled draft genomes from 38 human archaeological samples". I would not classify what they have as draft genomes, rather they have assembled contigs. Their status as genomes is best presented as single read referenced-based mapping to *C. tetani*, with reporting of % of the genome represented at different fold-coverages. It seems this analysis was done, but the results are only presented in Figure S17, which shows coverage for certain genes only. Why not report on average genomic coverage in a supplementary table?

Line 45-46 (as well as 290-295, 440, 520-525 and 570): "two novel *Clostridium* species". There is too much confidence placed in this claim. I'm curious if taxonomic assignment of the reads that comprise the contigs for the putative novel species would show close homology to other taxa given the complexity of the microbial content common to ancient datasets.

Figure 5 has very small font, the libraries seem to be randomly distributed in the list, and it's difficult to find the ones that are featured in downstream analyses. Can these data be presented in a more accessible way? Further to this, Figure 1 in the main manuscript shows a damage plot for Augsburg, SLC, Pericues, Griezno, and Punta-Candelero, which are purportedly those bins with the "highest damage rates", though the adjacent human mtDNA shows a pattern of consistently more damage. As a consequence I'm curious why the authors report "a significant correlation between damage rates of acBin and corresponding human mtDNA" (line 215-216). In looking at Figure S6, it seems the correlation was assessed with inclusion of UDG and non-UDG data, where damage has been either removed or partially repaired. I advise the authors to remake this plot with removal of those data points (the green and blue datasets in their Fig. S6 plot) to better present the relationship between human mtDNA and clostridial damage. Looking at the data, the relationship between the two variables will change.

The phylogeny they present in Figure 2c (and Figure S9) includes SanganjaA2_tooth, though their heterozygosity analysis shows this to be one of the five problematic datasets. I see in lines 190 – 193 the authors relieve concern over this by stating that strain variation is low enough for inclusion, but I also find it problematic that the two Sanganja and Tenerife, and also those from the Chinchorro/Chincha Bins group within modern diversity and show no branch shortening, which would be expected if their provenance were genuinely historical. Do the authors have an explanation for this trend? The Abusir genomes follow this expectation better.

Lines 180 – 190: What reference was used in the quality control of the acBins? The average reader (myself included) may not be familiar with CheckM, and this is an important authentication step that is worthy of greater explanation in the main manuscript.

Reviewer comments:

Line 43: “We assembled draft genomes from 38 human archaeological samples”. I would not classify what they have as draft genomes, rather they have assembled contigs. Their status as genomes is best presented as single read referenced-based mapping to *C. tetani*, with reporting of % of the genome represented at different fold-coverages. It seems this analysis was done, but the results are only presented in Figure S17, which shows coverage for certain genes only. Why not report on average genomic coverage in a supplementary table?

The genome coverage data is already in Supplementary Table S17, referenced on page 12. This data for 1X and 5X coverage has now also been added to Supplementary Table 2 (and referred to on page 6), to make this data more clear in the context of genome completion.

Line 45-46 (as well as 290-295, 440, 520-525 and 570): “two novel *Clostridium* species”. There is too much confidence placed in this claim. I’m curious if taxonomic assignment of the reads that comprise the contigs for the putative novel species would show close homology to other taxa given the complexity of the microbial content common to ancient datasets.

We have softened the wording of these instances of “novel *Clostridium*” to the editor-suggested wording used in the abstract (“new *Clostridium* lineages”).

Figure 5 has very small font, the libraries seem to be randomly distributed in the list, and it’s difficult to find the ones that are featured in downstream analyses. Can these data be presented in a more accessible way? Further to this, Figure 1 in the main manuscript shows a damage plot for Augsburg, SLC, Pericues, Griezno, and Punta-Candelerio, which are purportedly those bins with the “highest damage rates”, though the adjacent human mtDNA shows a pattern of consistently more damage. As a consequence I’m curious why the authors report “a significant correlation between damage rates of acBin and corresponding human mtDNA” (line 215-216). In looking at Figure S6, it seems the correlation was assessed with inclusion of UDG and non-UDG data, where damage has been either removed or partially repaired. I advise the authors to remake this plot with removal of those data points (the green and blue datasets in their Fig. S6 plot) to better present the relationship between human mtDNA and clostridial damage. Looking at the data, the relationship between the two variables will change.

There is no figure 5 in our paper and assume the reviewer is referring to Supplementary Figure 5.

We agree that there is too much being packed into a single figure. To address this, we have re-worked Supplementary Figure 5 with improved organization and larger font sizes, and have also ordered the samples by acBin damage rates. Additionally, we created a FigShare repository (<https://doi.org/10.6084/m9.figshare.23804106>) where readers can explore the mapDamage results as separate PDF files.

We have also remade Supplementary Figure 6 with the UDG datasets. The correlation only changed slightly and is still significant as before ($r = 0.62$, $R^2 = 0.38$, $p = 0.0028$) which doesn't change any conclusions. We have updated these values in the manuscript.

The phylogeny they present in Figure 2c (and Figure S9) includes SanganjiA2_tooth, though their heterozygosity analysis shows this to be one of the five problematic datasets. I see in lines 190 – 193 the authors relieve concern over this by stating that strain variation is low enough for inclusion, but I also find it problematic that the two Sanganji and Tenerife, and also those from the Chinchorro/Chincha Bins group within modern diversity and show no branch shortening, which would be expected if their provenance were genuinely historical. Do the authors have an explanation for this trend? The Abusir genomes follow this expectation better.

Yes, we do have a possible explanation. All of the samples the reviewer mentioned have relatively higher levels of strain variation or have low damage rates potentially indicative of being strains of more recent origin. Both of these reasons could contribute to longer branch lengths in the phylogeny. We have added the following text to clarify this:

“Notably, acBins from the Sanganji, Tenerife, Chinchorro, and Chincha samples do not show evidence of branch shortening in the tree indicative of ancient genomes, and instead cluster with modern strains. These acBins tend to have higher rates of strain variation, which could affect branch lengths, or low damage rates potentially indicative of a more recent origin (Supplementary Data 2).”

Lines 180 – 190: What reference was used in the quality control of the acBins? The average reader (myself included) may not be familiar with CheckM, and this is an important authentication step that is worthy of greater explanation in the main manuscript.

CheckM is a standard genome quality assessment tool used in metagenomics. It is also now the method used by the NCBI to assess genome quality.

CheckM estimates the genome completeness based on the detected presence of taxon-specific marker genes, and uses duplicated marker genes to estimate contamination and heterogeneity. We have added a statement describing CheckM in this section to clarify the method.